# Contributions of mirror-image hair cell orientation to mouse otolith organ and zebrafish neuromast function

Kazuya Ono[1†], Amandine Jarysta[2†], Natasha C Hughes[3†], Alma Jukic[4†], Hui Ho Vanessa Chang[3], Michael R Deans[5,6], Ruth Anne Eatock[1*], Kathleen E Cullen[3,7,8,9*], Katie S Kindt[4*], Basile Tarchini[2,10*]

[1]Department of Neurobiology, University of Chicago, Chicago, United States; [2]The Jackson Laboratory, Bar Harbor, United States; [3]Department of Biomedical Engineering, Johns Hopkins University, Baltimore, United States; [4]Section on Sensory Cell Development and Function, National Institute on Deafness and Other Communication Disorders, National Institutes of Health, Bethesda, United States; [5]Department of Neurobiology, Spencer Fox Eccles School of Medicine, University of Utah, Salt Lake City, United States; [6]Department of Otolaryngology - Head & Neck Surgery, Spencer Fox Eccles School of Medicine at the University of Utah, Salt Lake City, United States; [7]Department of Otolaryngology-Head and Neck Surgery, Johns Hopkins University, Baltimore, United States; [8]Department of Neuroscience, Johns Hopkins University, Baltimore, United States; [9]Kavli Neuroscience Discovery Institute, Johns Hopkins University, Baltimore, United States; [10]Tufts University School of Medicine, Boston, United States

*For correspondence:
eatock@uchicago.edu (RAE);
kathleen.cullen@jhu.edu (KEC);
katie.kindt@nih.gov (KSK);
basile.tarchini@jax.org (BT)

[†]These authors contributed equally to this work

Competing interest: The authors declare that no competing interests exist.

## eLife Assessment

This **valuable** study provides **convincing** evidence that mutant hair cells with abnormal, reversed polarity of their hair bundles in mouse otolith organs retain wild-type localization, mechanoelectrical transduction and firing properties of their afferent innervation, leading to mild behavioral dysfunction. It thus demonstrates that the bimodal pattern of afferent nerve projections in this organ is not causally related to the bimodal distribution of hair-bundle orientations, as also confirmed in the zebrafish lateral line. The work will be of interest to scientists interested in the development and function of the vestibular system as well as in planar cell polarity.

**Abstract** Otolith organs in the inner ear and neuromasts in the fish lateral-line harbor two populations of hair cells oriented to detect stimuli in opposing directions. The underlying mechanism is highly conserved: the transcription factor EMX2 is regionally expressed in just one hair cell population and acts through the receptor GPR156 to reverse cell orientation relative to the other population. In mouse and zebrafish, loss of Emx2 results in sensory organs that harbor only one hair cell orientation and are not innervated properly. In zebrafish, Emx2 also confers hair cells with reduced mechanosensory properties. Here, we leverage mouse and zebrafish models lacking GPR156 to determine how detecting stimuli of opposing directions serves vestibular function, and whether GPR156 has other roles besides orienting hair cells. We find that otolith organs in *Gpr156* mouse mutants have normal zonal organization and normal type I-II hair cell distribution and mechanoelectrical transduction properties. In contrast, *gpr156* zebrafish mutants lack the smaller mechanically evoked signals that characterize Emx2-positive hair cells. Loss of GPR156 does not affect orientation-selectivity of afferents in mouse utricle or zebrafish neuromasts. Consistent with normal

otolith organ anatomy and afferent selectivity, *Gpr156* mutant mice do not show overt vestibular dysfunction. Instead, performance on two tests that engage otolith organs is significantly altered – swimming and off-vertical-axis rotation. We conclude that GPR156 relays hair cell orientation and transduction information downstream of EMX2, but not selectivity for direction-specific afferents. These results clarify how molecular mechanisms that confer bi-directionality to sensory organs contribute to function, from single hair cell physiology to animal behavior.

## Introduction

The vestibular system provides essential information about the head's motion and orientation relative to gravity and plays a vital role in everyday life. Fundamental gaps exist in our understanding of vestibular development and function. For example, we do not fully understand how the abrupt reversal of hair cell (HC) orientation in otolith organs affects vestibular function, and whether the reversal also impacts HC mechano-electrical transduction properties or afferent neuron contacts. We explore these questions further in mouse and in the zebrafish lateral line, a sensory system where external fluid flow is detected by neuromast organs that also feature HC reversal.

Vestibular HCs reside in five sensory organs per inner ear in human, mouse and larval zebrafish: three cristae ampullaris and two otolith (or macular) organs. Within each of these organs, HCs are precisely oriented to detect mechanical stimuli based on direction. HCs are sensitive to the proportion of a stimulus aligned with the mechanosensory hair bundle's orientation: a vector from the bundle's short edge to its tall (kinociliary) edge. In cristae, which detect angular head motions, HCs are oriented uniformly to detect fluid flow in the plane of the attached semicircular canal. In contrast, HCs in otolith organs are oriented to respond to a range of head positions and linear motions in an approximately horizontal (utricle) or vertical (saccule) plane (*Figure 1A*). Within these planes, HCs show two opposing orientations across a virtual 'line of polarity reversal' (LPR), a conserved anatomical feature in all otolith organs (*Denman-Johnson and Forge, 1999*; *Flock, 1964*; *Lindeman, 1969*). HCs are also aligned in two opposing orientations in fish and amphibian neuromasts in the lateral-line system. Orientation reversal confers on neighboring HCs of opposing orientations opposite responses to the same local stimulus: depolarization and increased afferent spike rate in one population, but hyperpolarization and decreased spike rate in the other. This functional prediction and its utility are well established in neuromasts as the ability to sensitively detect water movements in two directions (*Chitnis et al., 2012*; *López-Schier et al., 2004*). Beyond increasing the range of stimulus orientations detected, it remains unclear how HCs with opposing orientations and responses serve otolith organ function.

In the utricle, the LPR also segregates distinct zones, for example the lateral extrastriola (LES) from the more medial striola (*Figure 1A*). Zones are strikingly different regions found in all vestibular epithelia, with 'central' and 'peripheral' zones of cristae corresponding to striolar and extrastriolar zones of maculae, respectively. Afferents that innervate striolar HCs have irregular spike timing and adapting responses in contrast to afferents that innervate extrastriolar HCs which have more regular and tonic spikes (*Goldberg, 1991*; *Goldberg, 2000*). Striolar afferents transmit more information about the time course of translational self-motion, while extrastriolar afferents better discriminate between different static head orientations relative to gravity (*Jamali et al., 2019*). Correlated with these physiological differences between zones are differences in HCs and afferent morphologies. Both the striola and extrastriola are populated by type I HCs contacted by *calyx afferent endings* and type II HCs contacted by *bouton afferent endings*. Most individual afferents in both zones contact both type I and type II HCs ('*dimorphic afferents*'). However, pure-calyx and pure-bouton individual afferents are only found in the striola and extrastriola, respectively (*Eatock and Songer, 2011*; *Goldberg, 2000*). In amniotes, there is no obligatory relationship between HC orientation reversal and zones. While the utricular striola is entirely medial to the LPR, the saccular striola is bisected by the LPR (*Figure 1A*; *Deans, 2013*; *Li et al., 2008*; *Xue and Peterson, 2006*). Moreover, central and peripheral zones of cristae bear many anatomical and physiological similarities to the striolar and extrastriolar zones, but cristae have no LPR and no abrupt variation in HC orientation, as in the organ of Corti.

Interestingly, mammalian otolith afferents have two distinct anatomical properties that segregate with the LPR. First, individual afferents are maximally excited by stimuli of one orientation and not the opposite (*Fernandez et al., 1972*; *Goldberg et al., 1990*), indicating that individual afferents selectively contact HCs sharing the same general orientation, as also observed in the zebrafish lateral line

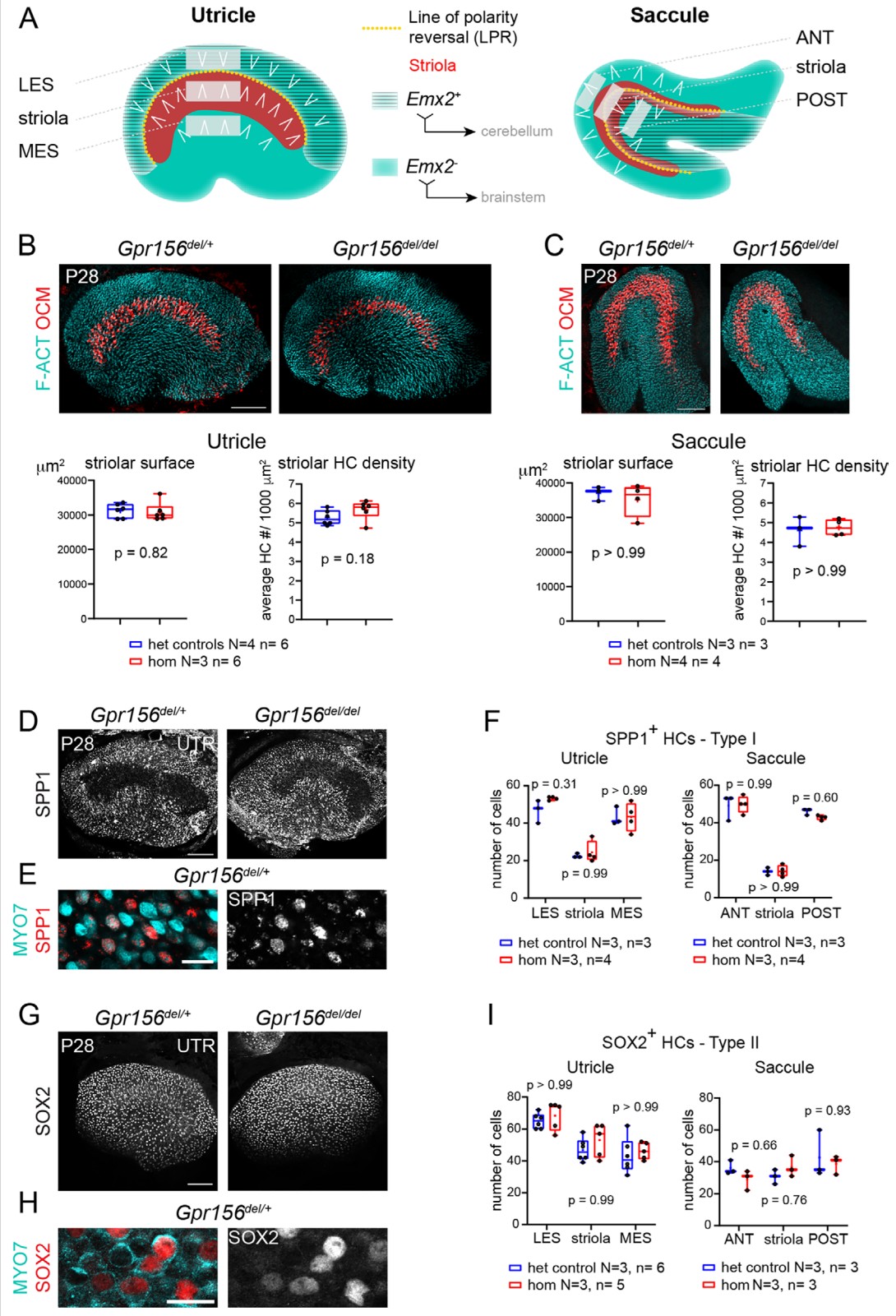

**Figure 1.** Normal striola and hair cell type organization in *Gpr156* mutants. (**A**) Diagrams showing general macular organization and the regions analyzed in (**B–I**) (LES, MES: lateral and medial extrastriola, respectively; ANT: anterior, POST: posterior). Each region is 130x50 μm in the utricle and 150x40 μm in the saccule. White chevrons indicate HC orientation. (**B–C**) P28 utricles (**B**) and saccules (**C**) immunolabeled with oncomodulin (OCM) to reveal the striolar region and phalloidin to reveal F-actin (F-ACT)-rich hair bundles. Graphs report striolar surface area and striolar HC density based

*Figure 1 continued on next page*

*Figure 1 continued*

on OCM labeling and show no change in mutants compared to heterozygote littermate controls. All points are graphed along with 25–75% boxplots (external bars: minimum and maximum, internal bar: median, cross: mean). Mann-Whitney U test. (**D–F**) P28 maculae immunolabeled with SPP1 to reveal type I HCs. Low magnification of the utricle (**D**), high magnification view showing SPP1 at the utricle HC neck (**E**), and quantification per region in utricle and saccule (**F**). (**G–I**) P28 maculae immunolabeled with SOX2 to reveal type II HCs. Low magnification of the utricle (**G**), high magnification view showing SOX2 in the utricle HC nucleus (**H**), and quantification per region in utricle and saccule (**I**). (**F**), (**I**): Two-way ANOVA with Sidak's multiple comparisons. N, n: number of animals and HCs, respectively. UTR, utricle. Scale bars: 100 μm (**B–C, D, G**), 10 μm (**E, H**).

(*Ji et al., 2018*). Second, the central projections of afferent neurons innervating HCs on each side of the LPR are also distinct; afferents that innervate one HC orientation largely project to the cerebellum, while the opposite HC orientation projects to the brainstem (*Ji et al., 2022*; *Maklad et al., 2010*). The LPR thus correlates with a striking segregation of both the peripheral and central projections of afferent neurons.

Recent work has illuminated how HCs with opposing orientations are formed during development (reviewed in *Tarchini, 2021*). Early in development all organs are polarized by core PCP proteins that are enriched asymmetrically at cell-cell junctions before HCs are born (*Deans, 2013*). In cristae, asymmetric core PCP proteins set up a default, uniform HC orientation across the epithelium. Because the asymmetric pattern of core PCP protein enrichment is unchanged across the LPR, a distinct, regional mechanism defines a HC population with an opposite orientation (*Deans, 2013*; *Deans et al., 2007*; *Jones et al., 2014*; *Mirkovic et al., 2012*). Regional expression of the transcription factor EMX2 in the lateral utricle and posterior saccule drives the 180°-reversed orientation of HCs compared to neighboring *Emx2*-negative HCs (*Holley et al., 2010*; *Jiang et al., 2017*). Although regionally restricted, *Emx2* is expressed by all cell types in the sensory epithelium. Reversal is achieved cell-autonomously in HCs by an orphan GPCR, GPR156 (*Kindt et al., 2021*). Unlike *Emx2*, *Gpr156* is transcribed throughout the sensory epithelia but is limited to HCs. In *Emx2*-positive HCs only, the GPR156 protein is trafficked and planar-polarized at apical junctions, where it signals through inhibitory G proteins to trigger a reversal in HC orientation. The EMX2 >GPR156 reversal cascade is conserved in zebrafish neuromasts and loss of either of these proteins produces neuromasts where all HCs adopt the same, uniform orientation (*Jacobo et al., 2019*; *Jiang et al., 2017*; *Kindt et al., 2021*; *Kozak et al., 2020*; *Lozano-Ortega et al., 2018*).

Importantly, in addition to reversing HC orientation, Emx2 is required for afferent selectivity. In zebrafish *emx2* mutants, individual afferent fibers either innervate nearly all or very few HCs instead of innervating ~50% of HCs based on their shared orientation (*Ji et al., 2018*). In mouse otolith organs, regional *Emx2* expression is required for local afferents that innervate HC of one orientation to project centrally to the cerebellum and not to the brainstem (*Ji et al., 2022*). In zebrafish, Emx2 was also recently shown to affect HC mechanotransduction, thus defining an orientation-based functional asymmetry in lateral-line neuromasts (*Chou et al., 2017*; *Kindig et al., 2023*). It remains unknown whether, similar to Emx2, GPR156 is required for orientation-based selectivity of afferent contacts in mouse otolith organs or zebrafish neuromasts, and whether Gpr156 participates in direction-based asymmetry of mechanosensitive response in neuromasts.

Here, using animal models where *Gpr156* is inactivated, we first established that abolishing the LPR in mouse otolith organs did not produce obvious changes in HC patterning and mechanotransduction properties. In contrast, we found that similar to Emx2, Gpr156 is required for asymmetric mechanotransduction in zebrafish neuromasts. Normal afferent segregation was maintained in both mice and zebrafish lacking GPR156 despite one HC population failing to reverse its orientation. Unchanged afferent organization in the mouse allowed us to specifically interrogate how the LPR serves vestibular function. We found that although *Gpr156* mutant mice did not show overt vestibular dysfunction, performance on two tests that engage otolith organs was significantly altered. This study clarifies how molecular mechanisms that orient HCs to achieve bi-directional sensitivity contribute to vestibular and lateral-line function.

## Results

### Normal hair cell numbers and organization in *Gpr156* mutant adult otolith organs

In previous work (*Kindt et al., 2021*), we established that in sensory organs of the inner ear and lateral-line, *Gpr156* expression is limited to HCs. In neonate otolith organs, we demonstrated that GPR156 is required for EMX2-positive HCs to adopt a reversed orientation compared to EMX2-negative HCs. We further showed that EMX2-positive HCs lacking GPR156 do not correct their orientation with time and remain non-reversed in young adults (P21-P28). Focusing on the utricular sensory epithelium, we showed normal macular surface area and number of HCs in all three zones (LES, striola and MES; *Figure 1A*) in adult *Gpr156^{del/del}* mutants. Here we begin by extending our morphological characterization of *Gpr156^{del/del}* adult macular organs to include the saccule as well as zonal and HC type profiles, before proceeding to functional and behavioral assessments.

We first verified that absence of GPR156 did not obviously interfere with striolar patterning in adult animals. We used oncomodulin (OCM) to immunolabel striolar HCs (*Simmons et al., 2010*) at 4 week of age (P28) and confirmed that striolar surface area and striolar HC density were unaffected in constitutive *Gpr156^{del/del}* mutant utricles and saccules compared to controls (*Gpr156^{del/+}*; *Figure 1B–C*). Each macular organ contains two main HC types, type I and type II. Therefore, we next quantified type I HCs in the LES, striola and MES zones of the adult utricle and in the anterior (ANT), striolar and posterior (POST) zones of the adult saccule. We used SPP1/Osteopontin, a marker that labels the neck region of approximately 90% of type I HCs (*Figure 1D–E*; *McInturff et al., 2018*). We observed no difference in SPP1-positive type I HC numbers between mutants and controls in any zone or organ (*Figure 1F*). We next used a SOX2 antibody to label type II HCs in the same zones (*Figure 1G–H*). Again, we observed no difference in type II HC numbers between mutants and controls (*Figure 1I*). In summary, loss of GPR156 prevents *Emx2*-positive HCs from reversing their orientation but does not affect macular patterning, including division into clear zones and appropriate numbers of type I and type II HCs throughout.

### No defects in mechano-electrical transduction were detected in individual mouse utricular hair cells of *Gpr156* mutants

*Gpr156* inactivation could be a powerful model to specifically ask how HC reversal contributes to vestibular function. However, GPR156 may have other confounding roles in HCs besides regulating their orientation, similar to EMX2, which impacts mechanotransduction in zebrafish HCs (*Kindig et al., 2023*) and afferent innervation in mouse and zebrafish HCs (*Ji et al., 2022*; *Ji et al., 2018*). After establishing normal numbers and types of mouse vestibular HCs, we thus first assessed whether HCs respond normally to hair bundle deflections in the absence of GPR156. For this assessment, we recorded whole-cell mechano-electrical transduction (MET) currents of HCs in the utricle of *Gpr156^{del/del}* mutants. To stimulate transduction, we deflected individual mechanosensory hair bundles of HCs from *Gpr156^{del/del}* mutants and heterozygous *Gpr156^{del/+}* control utricles with a rigid probe coupled to the back (short edge) of the bundle (see Methods, *Figure 2A*). All bundles were positively deflected along their orientation axes. $I_{MET}$ recordings were taken between P8 and P100 (median age for comparisons:~P20 for both null and heterozygous genotypes; see *Supplementary files 1–5*).

We first compared properties of the evoked MET current ($I_{MET}$) across genotypes in the LES, the zone in which hair bundles fail to reverse in *Gpr156* mutants.~70% (93/135) of LES HCs responded with detectable $I_{MET}$ (within 20 pA) to hair bundle deflection, regardless of HC type and genotype (Chi-squared test, *Supplementary file 1*). We attribute the insensitivity of the remaining cells to damage during dissection of the tissue, given that damaged bundles can be seen in every preparation, zone, and bundle type, independent of genotype.

$I_{MET}$ responses are shown for families of iterated 400 ms steps of displacement; subsets of these families are shown in *Figure 2B–E*. During bundle displacement, $I_{MET}$ typically peaked at step onset. As noted before (*Vollrath and Eatock, 2003*), traces evoked by stimuli in the middle of the operating range adapted with a time course fit by 1–2 decaying exponential terms. Because type I and II HCs were voltage-clamped at different holding potentials ($V_{hold}$ = –94 and –84 mV, respectively, see Methods), their MET currents experienced different driving forces. We normalized for driving force by converting current to conductance (G) and plotted G as a function of hair bundle displacement

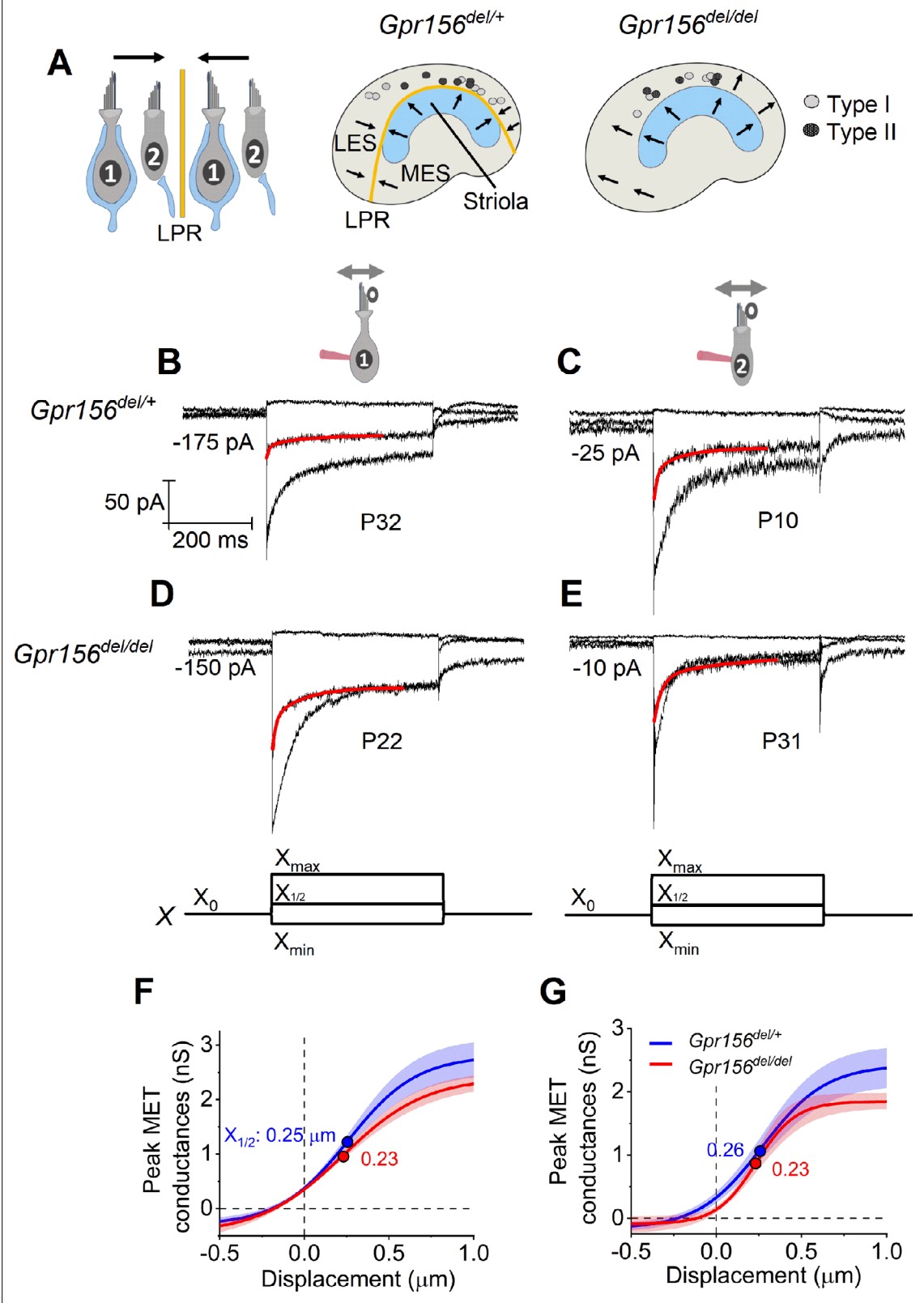

**Figure 2.** No effects of *Gpr156* deletion on mechanosensitivity were detected in LES hair cells. (**A**) *Left*, mal reversed bundle polarity at the LPR. *Arrows*, Directions of hair bundle motion that produce a depolarizing receptor potential. *Right,* Maps of recordings from LES zone, lateral to the LPR in heterozygotes, and similar location in null mutants, but without bundle reversal (arrows). (**B–E**) Exemplar $I_{MET}$ recordings from voltage-clamped LES HCs of different type (I vs. II, columns) and genotype (het vs. null, rows), and ranging in age from P10 to P31. XY scales for all 4 are in **D**. Each panel

*Figure 2 continued*

shows $I_{MET}$ current (average of 3 traces) evoked by 3 of many iterated bundle displacements (bottom of **D**, **E**): negatively and positively saturating stimuli ($X_{min}$, $X_{max}$) plus a step to ~ $X_{1/2}$ (midpoint) of the operating range. *Red traces*, fits of current decay for step to $X_{1/2}$ with **Equations 2 or 3** (Methods). Voltage was clamped with the indicated holding currents at –94 mV (type I) and –84 mV (type II). (**F,G**) *G(X)* relations from type I (**F**) and type II (**G**) HCs. G, was peak MET conductance taken at the onset of each of 20 or 40 displacement steps (**X**), iterated by 100 nm or 35 nm, respectively, and averaged across 3 repeated stimulus families. *Mean ± SEM; n=5–8 cells.* Fitted *G(X)* parameters are statistically similar for *Gpr156del/+* and *Gpr156del/del* HCs (see **Supplementary file 2**). *Filled circles,* $X_{1/2}$ values from fits. $X_{max}$: 1.1–1.3 µm. $X_{min}$; –0.2 to –0.45 µm.

(X) for each HC type and genotype (*Figure 2F–G*). G(X) curves were fit with a simple Boltzmann function (*Equation 1*, Methods), yielding values for maximum conductance ($G_{max}$), the midpoint of the G(X) curve ($X_{1/2}$, displacement at half-maximal $I_{MET}$) and *dx* (slope factor), the displacement range corresponding to an e-fold increase in G for small displacements. Among the LES HCs tested, no significant differences were detected between genotypes for any Boltzmann parameter ($G_{max}$, $X_{1/2}$, or *dx*) (*Figure 2F–G*, *Supplementary file 2*). $G_{max}$ values were in the range of 2.5–3.0 nS, $X_{1/2}$ was 200–300 nm and *dx* was 150–200 nm (Supp. File 2). From G(X) relations, we estimated the operating range (OR; displacement range yielding 10–90% growth of $G_{MET}$) to be ~800–900 nm for all LES HCs (Supp. File 2). In summary, the peak (non-adapted) displacement sensitivity of $I_{MET}$ was not detectably affected in LES hair bundles of mutants.

Another salient feature of vestibular HC transduction is adaptation (response decay) during a steady hair bundle deflection. We fit the adaptation with one or more exponential decays, and also calculated its extent (% decay; *Equation 4*, Methods; *Figure 2B–E*). As reported previously for immature mouse utricular HCs (*Vollrath and Eatock, 2003*), these mature mouse utricular HCs had fast and slow adaptation components. The fast component was more prominent at small displacements and the slow component dominated at large displacements; often, both were evident in the middle of the operating range. We therefore compared adaptation time constants, $\tau_{fast}$ and $\tau_{slow}$, at the midpoint of the operating range ($X_{1/2}$). For 400 ms steps, mean values are in the range of 4–10ms and 70–140ms, respectively, and the two components are responsible for comparable extents of the total adaptation. These results are consistent with previous reports with similar methods from immature mouse utricular HCs (*Vollrath and Eatock, 2003*) and immature rat saccular HCs (*Songer and Eatock, 2013*). In the current sample, however, adaptation at $X_{1/2}$ included a significant third, faster adaptation component ('very fast'; $\tau_{VF}$ < 1 ms) in most type I HCs (12/13) and 2/10 type II HCs. In the LES, we detected no significant differences with genotype in adaptation rates or extent (*Supplementary file 2*).

## No changes were detected in voltage-gated conductances of individual mouse utricular hair cells of *Gpr156* mutants

MET current initiates the receptor potential, which in turn modulates substantial voltage-gated currents in the HC's basolateral membrane, especially K+ currents, which further shape the receptor potential. For example, type I HCs make relatively fast, small receptor potentials compared to type II HCs because type I HCs have many more K channels that are open around resting potential. To test for effects of *Gpr156* deletion on voltage-dependent currents, we applied voltage step protocols to whole-cell clamped type I and type II HCs in the LES of *Gpr156del/del* and *Gpr156del/+* utricles (*Figure 3A–D*).

The time course and voltage dependence of outwardly rectifying voltage-gated current from LES type I cells of *Gpr156del/del* utricles resembled those of *Gpr156del/+* utricles (*Figure 3A and C*). Most of the current in type I HCs (*Figure 3A and C*) flows through a low-voltage-activated K+ conductance ($g_{K,L}$) that dominates the type I HC voltage response. As described in Methods, we used the current data to generate conductance density vs. voltage (G/Cm - V) curves (*Figure 3E*) and fitted them with the Boltzmann function (*Equation 5*, Methods) to show how the open probability of $g_{K,L}$ varies with voltage. In both genotypes, approximately half of $g_{K,L}$ was activated at resting membrane potential, $V_{rest}$ (~–85 mV). As a consequence of their large K+ conductance at $V_{rest}$, type I cells had low input resistances ($R_{in}$ ~50 MΩ). As is typical of type I HCs, the maximal K+ conductance density ($G_{max}/C_m$), which is proportional to the maximal number of open channels per unit membrane area, was much larger than for type II HCs (*Figure 3E* vs. 3 F, note difference in conductance scale; Supp. File 3).

Similarly, we detected no effect of genotype on the voltage-dependent outward currents of LES type II HCs (*Figure 3B, D and F*). The currents activated positive to resting potential (~–72 mV) and

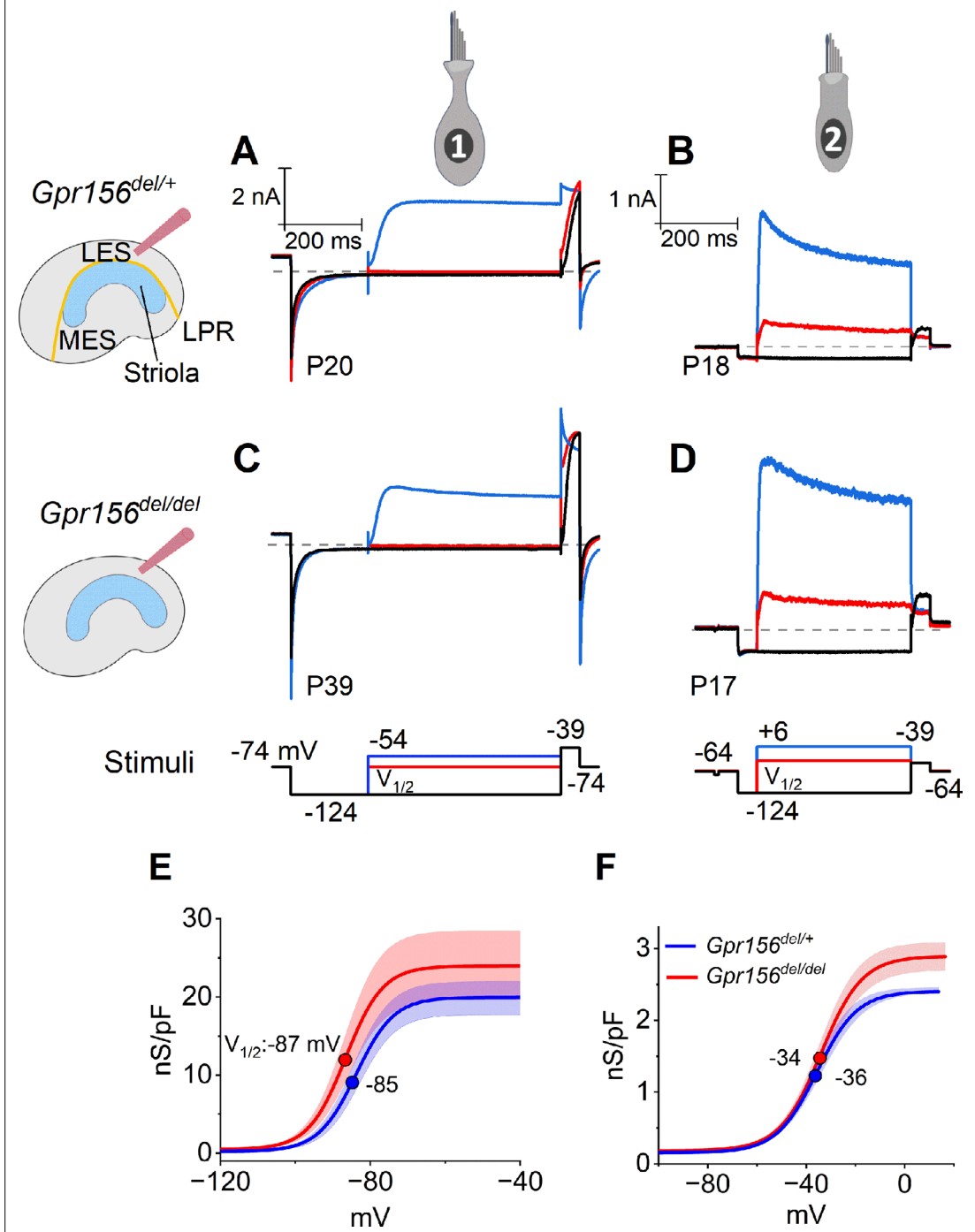

**Figure 3.** No effects of *Gpr156* deletion on basolateral voltage-dependent conductances were detected in LES HCs. (**A–D**) Exemplar voltage-dependent currents from lateral extrastriolar (LES) type I (**A, C**) and type II (**B, D**) HCs of *Gpr156*$^{del/+}$ (**A, B**) and *Gpr156*$^{del/del}$ (**C, D**) utricles. There are significant differences between HC types, but these differences are seen in both *Gpr156*$^{del/+}$ (**A, B**) and *Gpr156*$^{del/del}$ utricles (**C, D**). A pre-pulse from holding potential to –124 mV shows the usual type I-type II difference in K$^+$ conductances at or near resting potential: the prepulse deactivates $g_{K,L}$ in type I cells (**A,C**), and activates a small inwardly rectifying current in type II cells (**B, D**). Following the pre-pulse, test steps were iterated in 5 mV increments from –124 mV to –54 mV (type I) or from –124 mV to +6 mV (type II), activating currents with very different time course and voltage dependence. (**E, F**) G(V) relations measured from tail currents at the end of the iterated voltage steps show that $g_{K,L}$ activated positive to –114 mV (**E**) while type II delayed rectifying currents activated positive to –64 mV (**F**).

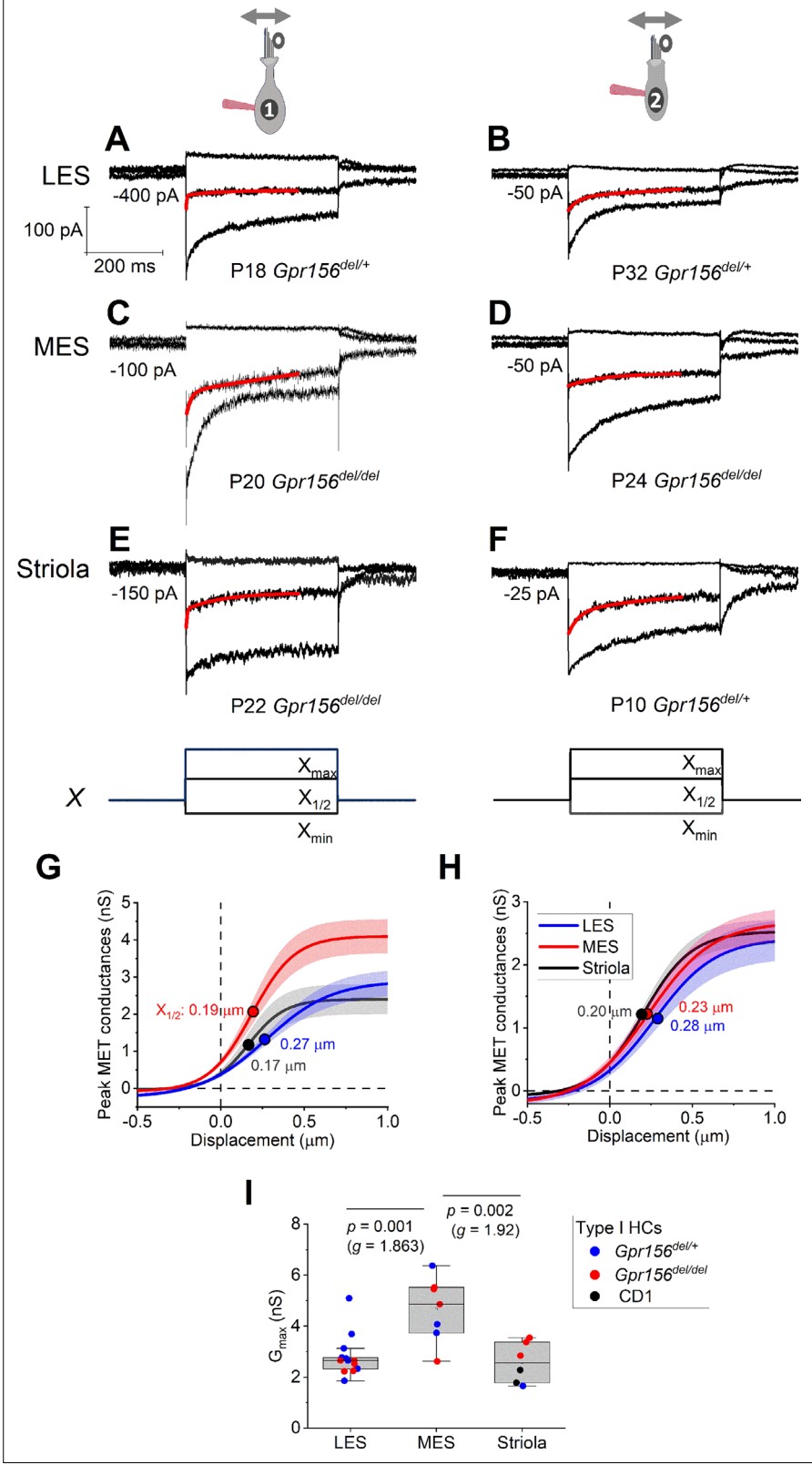

**Figure 4.** Displacement sensitivity and time course of step-evoked transduction currents in each zone were not strongly affected by Gpr156 deletion. (**A–F**) Exemplar $I_{MET}$ recordings from voltage-clamped HCs of different genotype, HC type (columns), and epithelial zone (rows). $I_{MET}$ current families (averages of 3) evoked by 3 of many iterated bundle displacements (**E**, **F**), bottom: negatively and positively saturating stimuli plus the step closest to

*Figure 4 continued on next page*

*Figure 4 continued*

the midpoint ($X_{1/2}$) of the G(X) relation. $X_{max}$: 1.08–1.32 µm. $X_{min}$: –0.2 to –0.45 µm. *Red traces*, fits of current decay with *Equations 2 or 3* (Methods). Holding potentials, –94 mV (type I) and –84 mV (type II). (**G–H**) Averaged peak (onset) $G_{MET}(X)$ relations for type I (**G**) and type II (**H**) HCs. Data from 5 to 9 HCs of each type were fitted with the Boltzmann functions, which were averaged to produce mean curves shown (± SEM). $X_{1/2}$ values (filled circles) are given. **I** Mean maximum $G_{MET}$ of type I HCs was larger in MES than in LES or striola, but there was not a significant effect of genotype. *Black circles*, 2 striolar type I HCs from CD1 mice.

had activation midpoints ($V_{1/2}$ values) of –30 to –40 mV. Because of the small voltage-gated conductance at rest, input resistance was ~10-fold higher than in type I HCs (~500 MΩ vs. ~50 MΩ). The outward currents inactivated significantly within 400ms, indicating a substantial A-current component. The maximum conductance density ($G_{max}/C_m$) at 400ms (~steady state), which is proportional to the number of open channels per unit membrane area, was much smaller than in type I HCs. Again, no effect of genotype was detected on the differences between type I and II HCs of the LES (*Supplementary file 3*).

Together these results are consistent with previous reports on key physiological properties of type I and II HCs in amniote vestibular organs. The mature physiological status of each HC type appeared to be preserved in LES HCs that failed to reverse their orientation because GPR156 was inactivated.

## Hair cell physiology beyond the lateral extrastriolar domain: Variations in properties were not related to loss of GPR156

Loss of GPR156 does not impact HC orientation in the striola and MES, but *Gpr156* is transcribed in all HCs across the maculae (*Kindt et al., 2021*). We thus investigated transduction and adaptation in those zones as well (*Figure 4*). While there was heterogeneity across HCs, we saw no striking systematic differences with genotype in the voltage-gated or mechanotransduction currents from control *vs. Gpr156* mutants (but note that the statistical power of the comparisons is low, *Supplementary file 4*).

## Gpr156 impacted the mechanosensitive properties of lateral-line hair cells in zebrafish

Similar to otolith organs of mammals, each neuromast within the posterior lateral-line system of zebrafish has two populations of HCs that enable the detection of anterior- or posterior-directed fluid flow (*Figure 5A*). To allow bi-directional detection, both Emx2 and Gpr156 are required to reverse the orientation of one HC population so that it can detect posterior flow (*Figure 5A*; *Jiang et al., 2017*; *Kindt et al., 2021*). Previous work has shown that in wild-type animals, the mechanosensitive properties of HCs that detect anterior flow (Emx2⁻) are larger compared to those that detect posterior flow (Emx2⁺; *Kindig et al., 2023*). Further, this work demonstrated that loss or gain of Emx2 in all lateral-line HCs was able to not only alter HC orientation, but also increase and decrease mechanosensitive responses, respectively (*Kindig et al., 2023*). Our current results in the mouse utricle did not detect significant differences in the mechanotransduction properties of HCs lacking GPR156 (*Figure 2*). Whether loss of Gpr156 alters the mechanosensitive properties of lateral-line HCs is not known.

To address this question, we used a membrane-localized GCaMP6s reporter in HCs (memG-CaMP6s; *Kindt et al., 2021*) to perform in vivo calcium imaging (*Lozano-Ortega et al., 2018*) and by imaging GCaMP6s signals in an apical plane containing the hair bundles, we measured evoked, mechanosensation-dependent calcium signals during fluid-jet stimulation. We used this approach to stimulate in both the anterior and posterior direction and compared the magnitude of the mechanosensitive responses in lateral-line HCs in both control (*gpr156⁺/⁻*) and *gpr156* mutants. As shown previously, we could reliably identify the orientation of control HCs (whether they sense posterior to anterior flow, P to A) or the reverse, A to P using this approach (e.g. *Figure 5B*). We first verified that in control animals, the magnitude of the mechanosensitive responses of HCs that detect anterior flow was greater than those that detect posterior flow (e.g.: *Figure 5B* and *Figure 5D and F*). Consistent with previous work, in *gpr156* mutants, we found that the majority of HCs only responded to P to A flow (e.g. *Figure 5C*; *Kindt et al., 2021*). We next examined the magnitude of the GCaMP6s response in these P to A HCs in *gpr156* mutants. If Gpr156 does not function downstream of Emx2 to alter the mechanosensitive properties of HCs, we predicted that the response magnitude of these HCs would fall between that of P to A and A to P control HCs. However, if Gpr156 does act downstream of Emx2,

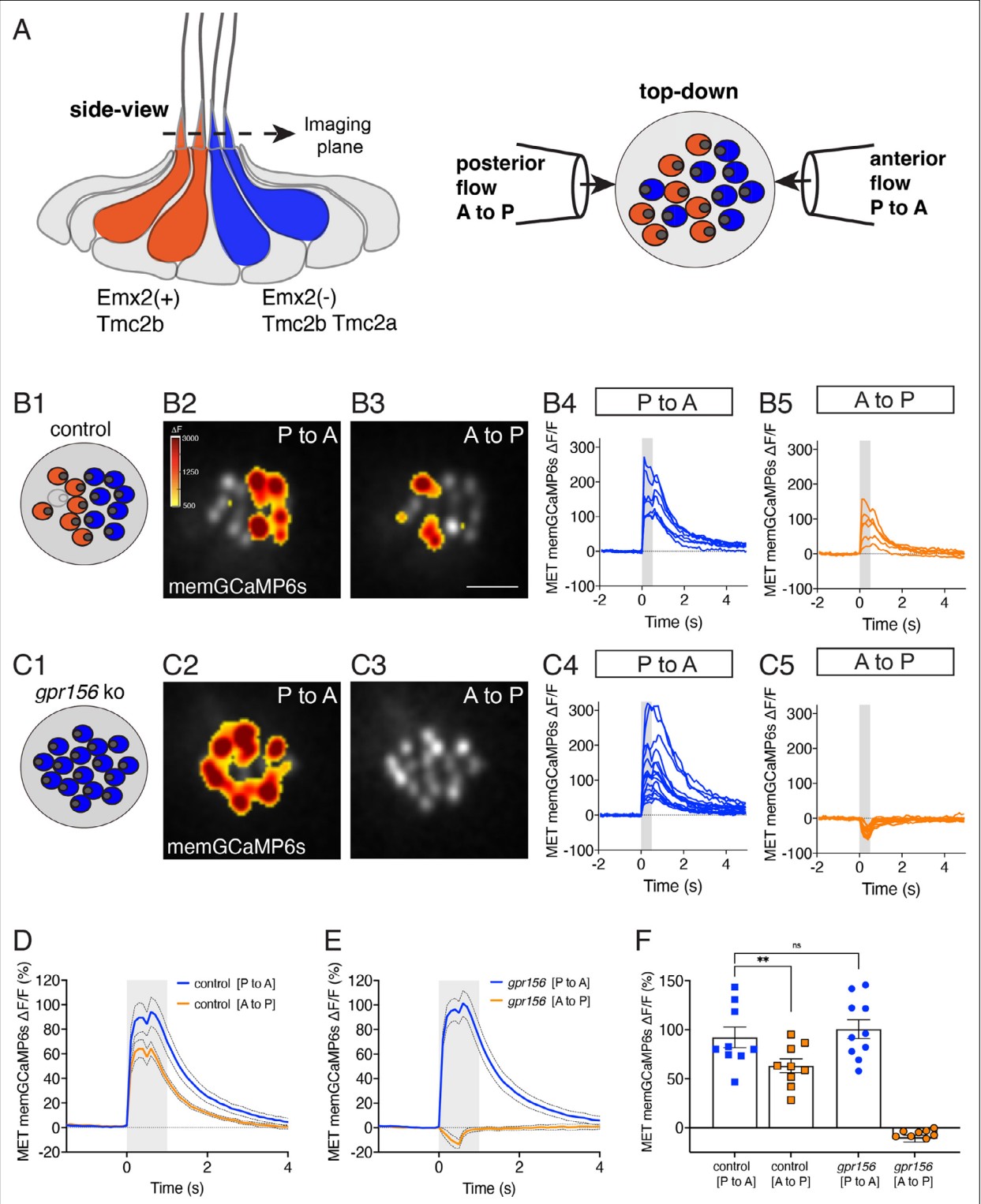

**Figure 5.** Gpr156 impacted the mechanosensitive properties of lateral-line HCs. (**A**) Schematic on the left shows a lateral-line neuromast from the side. HCs that detect anterior (P to A) and posterior (A to P) flow are color coded in blue and orange respectively. Posterior-sensitive HCs do not express Emx2 and rely primarily on the MET channel subunits Tmc2a and Tmc2b. In contrast, anterior-sensitive HCs express Emx2 and rely primary on Tmc2b for mechanosensation. Schematic on the right shows a plane taken through the apical hair bundle plane, viewed top-down. The directional sensitivity of each HC is dictated by the location of the kinocilium, which is indicated by the circle on the side of each hair bundle. This apical plane is one used to monitor mechanosensitive-calcium signals in lateral-line hair bundles. The pipettes on each side of the hair bundles show the direction of flow delivered

*Figure 5 continued on next page*

Figure 5 continued

to stimulate the two populations of HCs. (**B1–B5**) Representative example of evoked-mechanosensitive calcium signals in hair bundles of a control neuromast during a 500 ms anterior (**B2**) or a 500 ms posterior (**B3**) directed stimulus. Spatial patterns of GCaMP6s signals during stimulation (**B2–B3**) are colorized according to the ΔF heat map and superimposed onto a baseline GCaMP6s image. In **B1**, the colored circles indicate the respective hair bundle orientation observed from calcium imaging (A to P, orange; P to A, blue; no response, gray). ROIs were placed on each hair bundle to generate the ΔF/F GCaMP6s traces from individual hair bundles responding to P to A (**B4**) and A to P (**B5**) directed flow. (**C1–C5**) Representative example of evoked-mechanosensitive calcium signals in hair bundles of a *gpr156* mutant neuromast during a 500 ms P to A (**C2**) or a 500 ms A to P (**C3**) directed stimulus. Spatial patterns of GCaMP6s signals during stimulation (**C2–C3**) are colorized according to the ΔF heat map in **B2** and superimposed onto a baseline GCaMP6s image. In **C1**, the colored circles indicate the respective hair bundle orientation observed from calcium imaging (P to A, blue). ROIs were placed on all hair bundle to generate the ΔF/F GCaMP6s traces from individual hair bundles during P to A (**C4**) and A to P (**C5**) directed stimuli. (**D–F**) Quantification of the average increase in GCaMP6s per neuromast for P to A and A to P responding hair bundles. Traces in **D** and **E** show the average GCaMP6s response per neuromast in P to A and A to P hair-bundle populations (mean ± SEM is shown in **D–E**, n=9 control and 10 *gpr156* neuromasts). The magnitude of the GCaMP6s data in **D–E** is plotted to compare the average GCaMP6s increase for P to A and A to P hair bundles for each neuromast in **F**. In control neuromasts, the GCaMP6s increase in the hair bundles responding to P to A flow was larger compared to those responding to A to P flow. In *gpr156* mutants the GCaMP6s increase in hair bundles responding to P to A flow was significantly larger than control A to P cells, but not different than control P to A cells. **\*\*p=0.0043 and p=0.561, paired and unpaired t-test respectively. Scale bar is 5 µm in **B3**.

we predicted that the response magnitude of the P to A HCs in *gpr156* mutants would resemble that of P to A HCs in controls and be greater than that of A to P HCs in controls. Importantly, we found that the magnitude of the mechanosensitive responses in *gpr156* mutants was comparable to that of control P to A HCs (e.g., *Figure 5C* and *Figure 5E–F*). Further, the mechanosensitive responses were significantly larger in *gpr156* mutants than control A to P HCs (*Figure 5F*).

Our analysis thus indicates that in *gpr156* mutants, lateral-line HCs show larger mechanosensitive responses. These responses are similar to HCs in control animals that respond to anterior flow (Emx2⁻). Together, this data suggests that the EMX2-GPR156 mechanism driving HC orientation reversal also affects the mechanosensitive properties of HCs in zebrafish neuromasts.

## Segregated afferent receptive fields were preserved in mouse utricles lacking GPR156

After characterizing the functional properties of HCs in mouse and zebrafish *Gpr156* mutants, we examined whether GPR156 plays a role in orientation-selective afferent innervation. Previous work has demonstrated that *Emx2* expression in the LES of the mouse utricle and in HCs that sense posterior flow in the lateral line is required for afferents to innervate HCs of similar (rather than opposing) orientation (*Ji et al., 2022*; *Ji et al., 2018*). Whether GPR156 plays a similar role in direction-selective afferent innervation is unknown.

To investigate whether afferent organization in the LES is disrupted by loss of GPR156, we first compared the terminal fields of individual dye-filled calyx-bearing afferents in *Gpr156*^del/+^ and *Gpr156*^del/del^ utricles. All LES calyces belong to dimorphic afferents, which form both calyces on type I HCs and boutons on type II HCs. To examine terminal fields, a fluorescent internal solution was injected into a calyx terminal within the LES during whole-cell recordings. The fluorescent dye was then allowed to diffuse into the calyx and from there throughout the terminal arbor.

Using this approach, we found that terminal arbors (receptive fields) in *Gpr156*^del/+^ control utricles (n=12) comprised 1 or 2 calyces and as many as 65 boutons (median: 29; *Figure 6A–C*). The labeled afferent terminals were restricted to LES in all but one case (*Figure 6A*). In this single case, a terminal branch crossed the LPR, innervating 16 striolar HCs with opposite orientation to the 12 LES HCs also innervated. In some filled LES afferents, a thin branch originated below the epithelium (*Figure 6C*, *Figure 6—video 1*), as previously described for chinchilla extrastriola (*Fernández et al., 1990*). We also labeled a single striolar calyx from a control utricle near the striolar/MES boundary; its terminal field was excluded from the calbindin-negative MES (*Figure 6D*).

When we performed dye labeling in *Gpr156*^del/del^ utricles, the innervation patterns and the size of terminal fields appeared similar to *Gpr156*^del/+^ controls. In *Gpr156*^del/del^ utricles, the labeled LES afferents (n=8) were also all dimorphic, terminating in up to three calyces and 56 boutons (median: 23). The labeled receptive fields did not cross the striola/LES border (*Figure 6E–G*). We also did not observe fiber branches extending across the border from parent axons (*Figure 6E–G*). As observed in the lone *Gpr156*^del/+^ striolar afferent (*Figure 6D*), *Gpr156*^del/del^ afferents with the main dendritic fiber

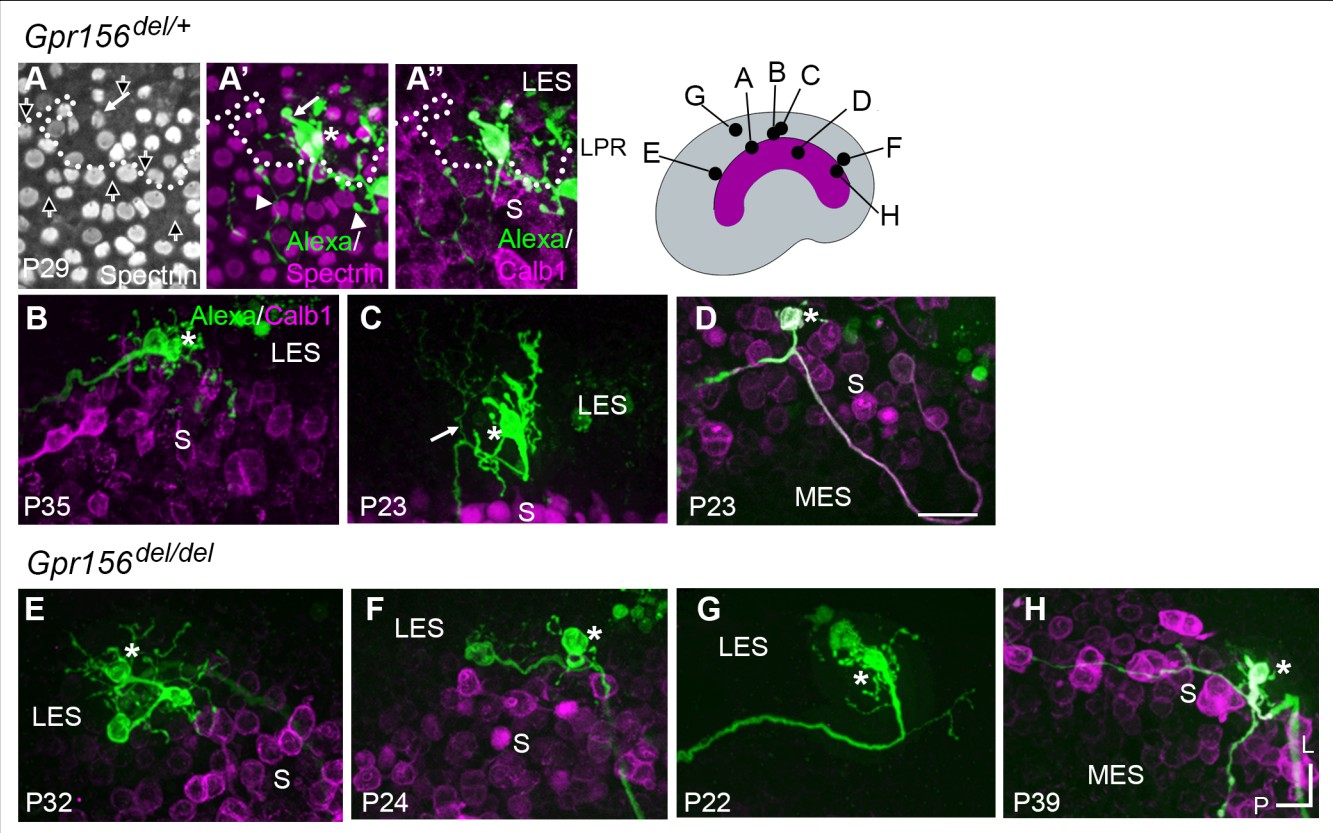

**Figure 6.** Afferent innervation patterns near the LES/S zone boundary were not substantially disturbed by *Gpr156* deletion. Afferent receptive fields (*green*) were labeled by diffusion of fluorescent dye (AlexaFluor) from whole-cell recording pipettes into calyces (*asterisks*) and throughout the terminal arbor, for (**A–D**) *Gpr156del/+* controls and (**E–H**) *Gpr156del/del* mutants. Counterstained with anti-calbindin (Calb1) antibody to show the striola (*magenta*). *Top right,* Schematic of the utricle with magenta striola; *black dots,* approximate location of each labelled calyx *shown*. All labeled afferents had a thick, medial-projecting neurite that branched to form up to two calyces and many bouton contacts. Anti-βII-spectrin labeling (**A, A'**) leaves an unlabeled hole where the kinocilium is, allowing determination of bundle orientation *black arrows outlined in white,* (**A**) and, in *Gpr156del/+* controls, the LPR (*dotted white line*). (**A**) In one control afferent, the receptive field straddled the LPR (**A', A''**), with 1 calyx on a type I HC in the LES (*white arrow*) and some boutons contacting type II HCs, as terminals or en passant, in the calbindin+ striola (*white arrowheads*). (**B, C**). In all other fills, the labeled LES arbors innervated only LES HCs. (**C**) *Arrow,* A thin branch extended from the fiber below the epithelium. (**D**) A receptive field labeled by filling a striolar calyx included 2 calyces and some bouton endings, all restricted to the calbindin+ striola (this afferent is white because of the merge of AlexaFluor and calbindin stains). (**E–G**) Afferent terminal fields of LES calyces from *Gpr156del/del* utricles largely remained in the calbindin− region (LES) (**E**), *Figure 6— video 2*. (**H**) A striolar (calbindin+) calyx in a *Gpr156del/del* mouse made multiple boutons entirely in the calbindin+ area (striola). Scale bar: 20 μm L, lateral; P, posterior.

The online version of this article includes the following video(s) for figure 6:

**Figure 6—video 1.** Confocal stack volume illustrating afferent terminal arbors (receptive fields) in Gpr156del/+ control utricles (see *Figure 6C*).
https://elifesciences.org/articles/97674/figures#fig6video1

**Figure 6—video 2.** Confocal stack volume illustrating afferent terminal arbors (receptive fields) in Gpr156del/del mutant utricles (see *Figure 6E*).
https://elifesciences.org/articles/97674/figures#fig6video2

terminating in the striola (n=4) contacted relatively few HCs, and receptive fields were restricted to the calbindin-positive region (*Figure 6H*).

To confirm and complement dye-filling results at the whole organ level, we used a new genetic approach to specifically label afferents innervating EMX2+ HCs in the LES. Interestingly, when breeding the *Advillin-Cre* strain (*Zhou et al., 2010*) with the *Tigre Ai140* reporter (*Daigle et al., 2018*), we only observed Cre recombination and EGFP expression in vestibular ganglion neurons that contacted HCs in the lateral utricle region (*Figure 7A*). Close examination revealed clear labeling of HC calyces in the LES, and occasionally in OCM+ HCs in the lateral striola (*Figure 7B*, arrowheads). We confirmed that similar to OCM− EGFP+ HCs in the LES, these OCM+ EGFP+ HCs were invariably oriented towards the medial edge (*Figure 7C*). This demonstrated that *Advillin* expression defines a population of neurons

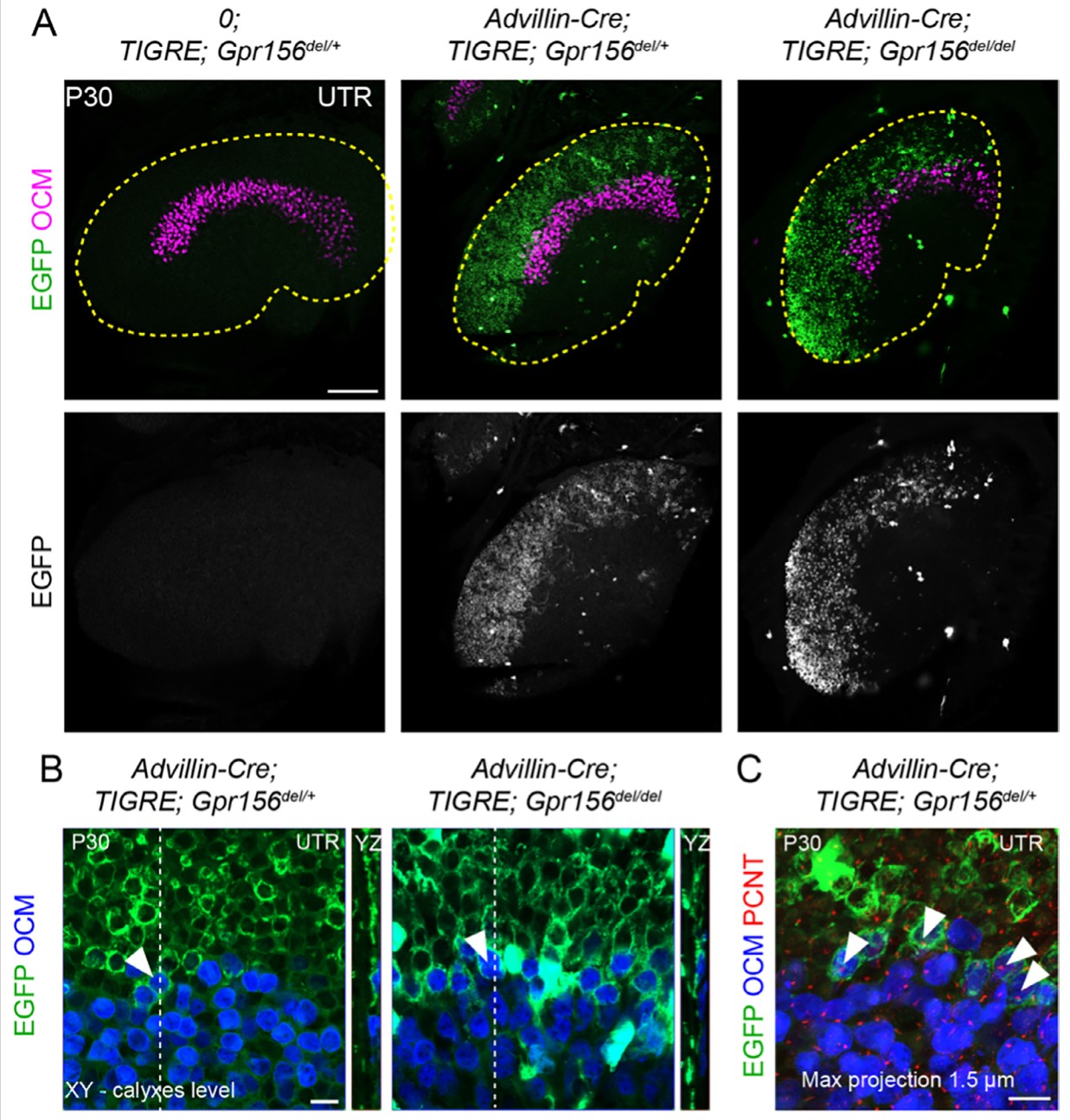

**Figure 7.** LES afferent innervation pattern was comparable for *Gpr156*<sup>del/+</sup> and *Gpr156*<sup>del/del</sup> utricles. (**A**) P30 utricles where oncomodulin (OCM) labels striolar HCs and *Advillin*-positive afferents are labeled by EGFP in *Advillin-Cre; TIGRE; Gpr156* animals. *Advillin-Cre* neurons specifically innervated lateral HCs and orientation-based segregation was not altered in *Gpr156*<sup>del/del</sup> mutants. (**B**) Close-up views at the LES-striola border (XY and YZ views). The vast majority of HCs innervated by EGFP-positive neurons were in the the LES but *Advillin*-positive neurons also innervated OCM⁺ HCs in the lateral striola in both control and *Gpr156* mutants. (**C**) Close-up view where basal bodies are labeled with pericentrin (PCNT) to reveal HC orientation (arrowheads). Lateral striolar HCs innervated by *Advillin* neurons were consistently oriented medially, showing that innervation from *Advillin* neurons strictly segregates with HC orientation, and not with striolar characteristics (OCM expression). Yellow dotted lines in A mark the outline of the utricle. Scale bars: 100 μm (**A**) 10 μm (**B–C**).

whose peripheral processes closely segregate with HC orientation, irrespective of striolar (OCM⁺) or extrastriolar (OCM⁻) HC identity. Importantly, HC innervation by *Advillin*-positive neurons was unchanged in *Gpr156*<sup>del/del</sup> mutants (**Figure 7A–B**). We observed equivalent numbers of OCM⁺ HCs innervated in the lateral striola in *Gpr156*<sup>del/del</sup> mutants compared to controls (Mann-Whitney p=0.74; average OCM⁺/EGFP⁺ HCs per utricle: *Gpr156*<sup>del/+</sup>: 20.25, *Gpr156*<sup>del/del</sup>: 19.25; three animals and four

utricles per genotype; no OCM$^+$/EGFP$^+$ HCs oriented towards the lateral edge were recorded in either genotype).

In summary, we detected no anomaly in afferent receptive fields in utricles lacking GPR156 and thus lacking a LPR, showing that the tendency of afferents to innervate HCs on one side of the LPR is not controlled by GPR156 or HC orientation.

## Afferent firing patterns were preserved in mouse utricles lacking GPR156

Our innervation data suggested that HCs in zones demarcated by the LPR are still innervated by distinct populations of afferent neurons in *Gpr156$^{del/del}$* mutants. We next investigated whether the LES afferent population retains normal physiology. Striolar afferents have highly irregular inter-spike intervals, while extrastriolar afferents in both LES and MES have much more regular inter-spike intervals (*Goldberg, 1991*; *Goldberg, 2000*; *Goldberg et al., 1990*). Associated with the difference in spike timing regularity are zonal differences in response dynamics and information encoding capacity (e.g. *Jamali et al., 2019*). The spike timing difference is considered to reflect differences in afferents' excitability as conferred by their intrinsic ion channels, with regular afferents being more excitable (*Kalluri et al., 2010*; *Ventura and Kalluri, 2019*). To test for effects of *Gpr156* inactivation on afferent excitability, we injected small depolarizing current steps into LES calyces through the whole-cell recording pipette and measured current threshold for spiking, I$_{thresh}$, and the numbers of spikes evoked by each current step.

As reported (*González-Garrido et al., 2021*; *Ono et al., 2020*), I$_{thresh}$ evoked 1-to-several spikes but increments beyond I$_{thresh}$ evoked more sustained firing (*Figure 8A–B*). No difference was observed across genotypes (*Supplementary file 5*): in both *Gpr156$^{del/+}$* afferents (n=18) and *Gpr156$^{del/del}$* afferents (n=20), mean I$_{thresh}$ was ~100 pA and ~95% of afferents produced sustained spiking at 3×I$_{thresh}$. These results suggest that afferent firing patterns, and likely the underlying ion channel expression, are unaltered in the LES of *Gpr156$^{del/del}$* utricles, despite abnormal HC orientation.

## Gpr156 was not required for afferent selectivity or synaptic pairing in zebrafish lateral-line

We next expanded our analysis to determine whether loss of Gpr156 impacts afferent selectivity in the lateral-line. For this analysis, we used mosaic expression of *neuroD:tdTomato* to label single lateral-line afferent neurons. After identifying *gpr156* mutant and control (*gpr156$^{+/-}$*) animals with labeled fibers, we immunolabeled to visualize orientation (phalloidin), HC bodies (MYO7), and cell type (presence or absence of Emx2). We then used confocal microscopy to image the innervation pattern of single afferents.

Consistent with previous results, some afferent fibers selectively contacted Emx2$^-$ HCs with hair bundles oriented to respond to anterior flow in control animals (*Figure 9A–H*). Other afferent fibers selectively contacted Emx2$^+$ HCs with hair bundles oriented to respond to posterior flow (*Figure 9—figure supplement 1A–H*). We then examined the innervation patterns of afferent fibers in *gpr156* mutants. Similar to controls we were able to identify fibers with a clear preference for Emx2$^-$ or Emx2$^+$ HCs (*Figure 9I–P*, *Figure 9—figure supplement 1I–P*). Importantly, in *gpr156* mutants, afferent fibers innervated HCs based on the presence or absence of Emx2, despite all HCs responding to anterior flow.

Following this initial assessment, we quantified the percentage of HCs per neuromast that were innervated by each afferent fiber. We found that *gpr156* mutant fibers innervated a similar percentage of HCs per neuromast compared to controls (*Figure 9Q*, control: 38.90% ± 2.40, n=18, *gpr156* ko: 41.31% ± 2.33, n=23, p=0.48). We also quantified how selective each afferent fiber was for Emx2$^+$ or Emx2$^-$ HC. For each afferent fiber, we calculated the percentage of either Emx2$^+$ or Emx2$^-$ HCs innervated out of the total number of innervated HCs. In both control and *gpr156* mutants, afferent fibers showed a high selectivity (*Figure 9R*, control: 85.06% ± 4.32, n=18, *gpr156* ko: 81.29% ± 3.17, n=23, p=0.48). We split this dataset and examined the selectivity of afferent fibers that preferentially contacted either Emx2$^-$ or Emx2$^+$ HCs (*Figure 9—figure supplement1S–T*). This analysis revealed that in control and *gpr156* mutants, both afferent fiber types were highly selective (afferents contacting Emx2$^-$ cells, control: 81.41% ± 7.90, n=8, *gpr156* ko: 79.37% ± 4.73, n=10, p=0.82; afferents contacting Emx2$^+$ cells, control: 87.99%±4.75, n=10, *gpr156* ko: 82.80% ± 4.38, n=13, p=0.43).

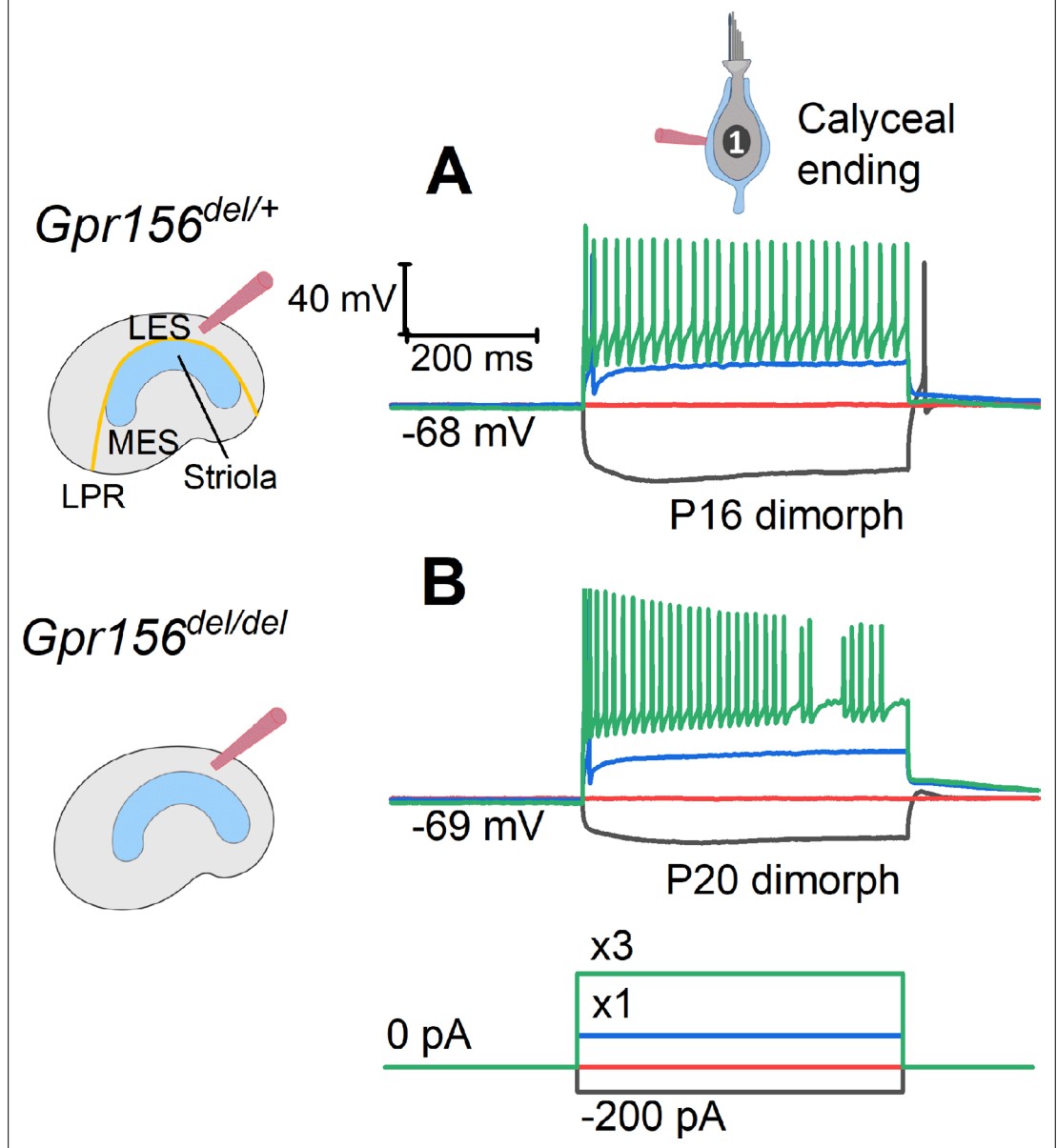

**Figure 8.** The excitability of LES afferents was not significantly affected by Gpr156 deletion. Exemplary voltage responses of control (**A**) and null (**B**) afferents to injected steps of current. We recorded from the large calyceal endings as a way to access these dimorphic afferents, which, like most LES afferents, made contact with both type I and type II HCs. For current steps (bottom) at 3×I$_{thresh}$ (×3, green), both genotypes produced sustained firing, considered typical of regular afferents innervating LES and MES. Here I$_{thresh}$ (blue trace) was 100 pA for Gpr156$^{del/+}$ afferent and 250 pA for Gpr156$^{del/del}$ afferent, but overall there was no significant difference in current threshold with genotype.

Overall, our single fiber labeling revealed that Gpr156 is not required for afferent fibers to selectively innervate HCs based on the presence or absence of Emx2.

To extend this analysis, we examined synaptic pairing in *gpr156* mutants. Previous research found that along with inappropriate innervation, there is an increase in the amount of unpaired pre- and post-synapses in *emx2* mutants (*Ji et al., 2018*). We immunostained control along with *gpr156* and *emx2* mutant HCs with Ribeye b to label presynapses and pan-MAGUK to label postsynapses (*Figure 10A–C*). We then quantified the number of complete immunolabeled synapses per HC. Among all genotypes, the number of complete synapses was unaltered (*Figure 10D*, control 3.56 ± 0.06, n=9 neuromasts, *gpr156* ko: 3.61 ± 0.10, n=16 neuromasts, *P*=0.74; control 3.51±0.27, n=9 neuromasts, *emx2* ko: 3.67±0.08 n=9 neuromasts, p=0.50). We also quantified unpaired pre- and post-synapses. Similar to previous results, there were more unpaired pre- and post-synapses per

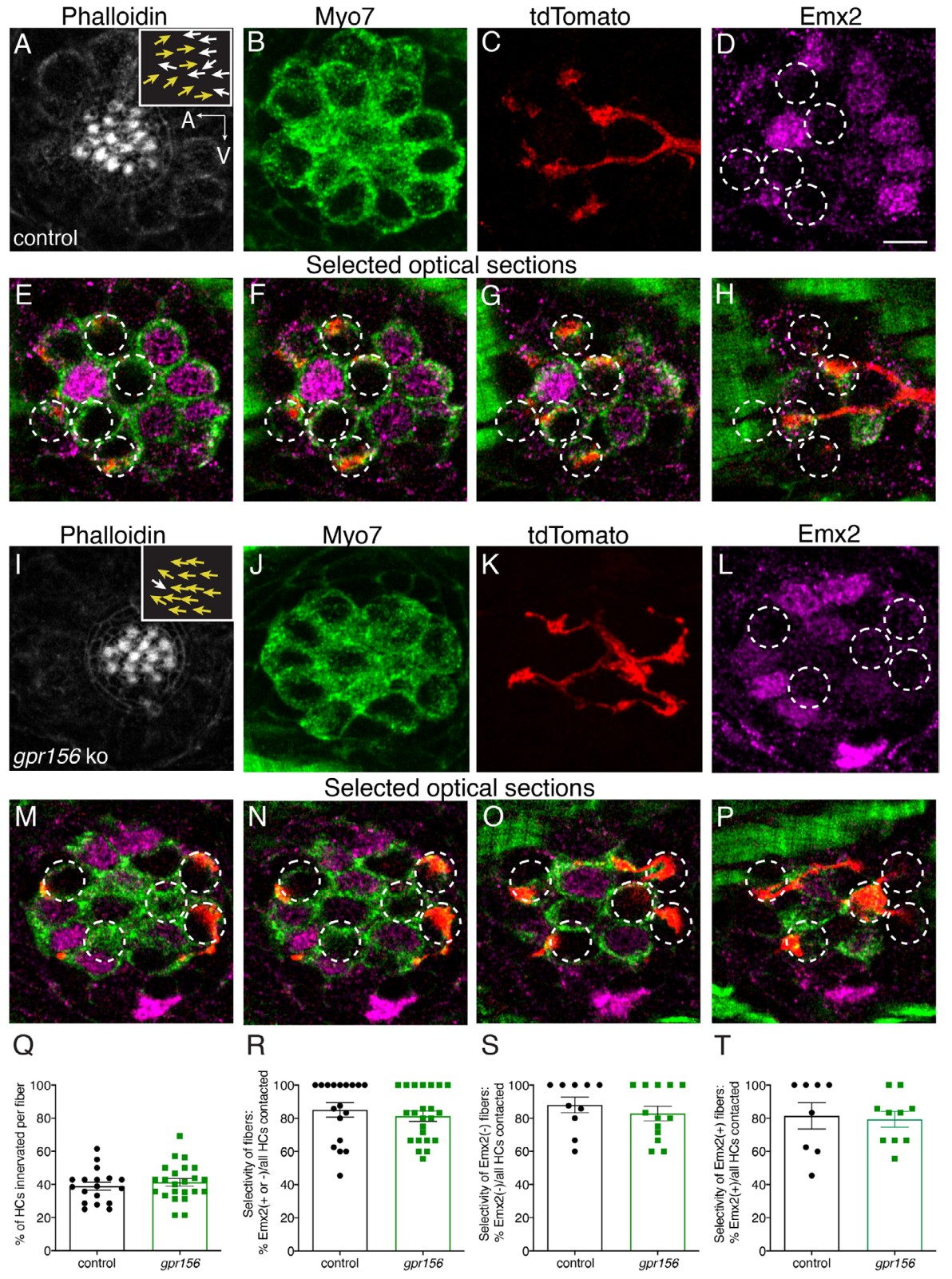

**Figure 9.** Gpr156 was not required for lateral-line afferents to select Emx2$^+$ or Emx2$^-$ hair cells. (**A–H**) Images of a control neuromast innervated by a single afferent fiber that contacts Emx2(-) cells at + dpf. (**A**) Phalloidin label reveals 16 hair bundles (8 A to P and 8 P to A hair bundles). Z-stack projections show all HCs labeled with Myo7a (**B**) a single afferent fiber expressing tdTomato (**C**) and Emx2(+and -) HCs (**D**). (**E–H**) Selected optical sections of (**B–D**) highlight the single afferent contacting individual Emx2(-) HCs. (**I–P**) Images of a *gpr156* mutant neuromast innervated by a single

*Figure 9 continued on next page*

*Figure 9 continued*

afferent fiber that contacts Emx2(-) cells. (**I**) Phalloidin label revealed 14 hair bundles (13 A to P and 1 P to A hair bundles). Z-stack projections show all HCs labeled with Myo7a (**J**) a single afferent fiber expressing tdTomato (**K**) and Emx2(+ and -) HCs (**L**). **M–O** Selected optical sections of (**J–L**) highlight a single afferent in a *gpr156* mutants contacting individual Emx2(-) HCs. (**Q**) In both controls and *gpr156* mutants, each afferent fiber contacted the same number of HCs per neuromast. **R** The overall selectivity of afferent fibers for Emx2(+ or -) HCs were similarly high in *gpr156* mutant and controls. (**S–T**) The selectivity of afferent fibers for Emx2(+) or Emx2(-) HCs was similarly high in both *gpr156* mutants and controls. Arrows in **A** and **I** indicate the orientation of the hair bundles in each example. SEM is shown in **Q-T**. An unpaired *t*-test was used for comparisons. Scale bar = 5 µm.

The online version of this article includes the following figure supplement(s) for figure 9:

**Figure supplement 1.** Gpr156 was not required for lateral-line afferents to select Emx2⁺ HCs.

neuromast in *emx2* mutants compared to controls (*Figure 9E–F*, presynapses, *emx2+/+* or *emx2+/-* control 2.00±0.53, n=9 neuromasts, *emx2* ko: 4.22±0.70, n=9 neuromasts, p=0.02; postsynapses, control 1.44 ± 0.56, n=9 neuromasts, *emx2* ko: 6.56±0.78, n=9 neuromasts, p<0.0001). In contrast, in *gpr156* mutants, there was no increase in unpaired pre- or post-synapses per neuromast compared to controls (*Figure 10E–F*, presynapses, *gpr156+/-* control 2.11±0.31, n=9 neuromasts, *gpr156* ko: 1.68±0.23, n=16 neuromasts, p=0.29; postsynapses, control 0.33±0.17, n=9 neuromasts, *gpr156* ko: 0.75±0.17, n=16 neuromasts, p=0.12). A lack of unpaired pre- and post-synapses in *gpr156* mutants further confirms that Emx2 but not Gpr156 impacts afferent innervation in lateral-line HCs.

In summary, *Gpr156* inactivation does not appear to impact afferent selectivity in either mouse or zebrafish. This is in sharp contrast with results in *Emx2* mutants, where orientation-specific HC afferent innervation is profoundly altered in mouse and zebrafish (*Ji et al., 2022*; *Ji et al., 2018*). It follows that HC orientation per se does not dictate afferent selectivity, with EMX2 likely using a different effector(s)

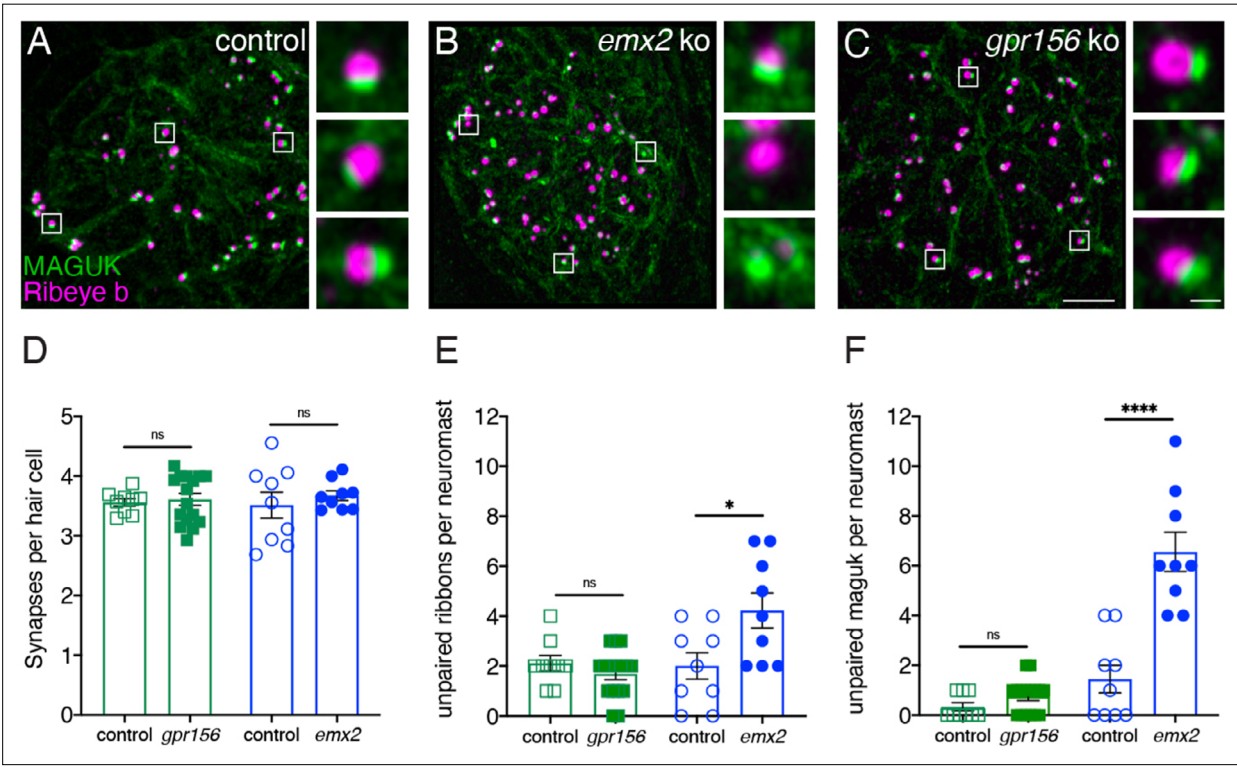

**Figure 10.** Grp156 was not required for pairing of pre- and post-synapses in lateral-line hair cells. (**A–C**) Images neuromasts immunolabeled with Maguk and Ribeye b to stain HC post- and pre-synapses, respectively at 5 dpf. (**A**) Example of an immunostain of a control neuromast showing three complete synapses to the right. (**B**) Image of an *emx2* mutant neuromast that shows complete synapses, as well as unpaired pre- and postsynapses, shown to the right. (**C**) Image of a *gpr156* mutant neuromast, with complete synapses shown to the right. (**D**) Quantification revealed the same number of complete synapses per HC in *emx2* and *gpr156* mutants compared to controls. (**E–F**) In *gpr156* mutants, quantification revealed there is no difference in the number of unpaired pre- or post-synapses per neuromast compared to controls. In *emx2* mutants, there were significantly more pre- or post-synapses per neuromast compared to controls. SEM is shown in **D–F**. Scale bar = 5 µm, 0.5 µm for insets. Unpaired t-test were used to make comparisons. *p<0.05, ****p<0.0001.

than GPR156 in this context. Thus, afferent studies confirm that the *Gpr156^del^* model is well-suited to investigate specifically how the reversal of HC orientation in otolith and lateral line organs serves organ function.

## *Gpr156* mutant mice had defects in otolith-driven, but not canal-driven, vestibular behaviors

In mammals, only otolith organs (utricle, saccule) have a LPR. Therefore, the absence of a LPR in *Gpr156* mutants should only affect otolith-driven reflexes and behaviors. In order to test this prediction we compared *Gpr156* control and mutant mice on a number of tests that can discriminate between otolith and canal inputs.

### Normal posture, balance beam but impaired swimming performance in *Gpr156* mutants

To assess how loss of the LPR affects vestibular functions, we first determined whether *Gpr156* mutant mice showed postural/balance impairments. Visual and quantitative assessments did not reveal circling or head tremor behaviors in *Gpr156^del/del^* mice at rest (*Figure 11—figure supplement 1*). Additionally, we conducted basic standard behavioral tests that involve subjective scoring: (1) tail hanging, (2) air righting, and (3) contact inhibition of righting. Both control and *Gpr156^del/del^* mice scored 0 (where 0=normal, see Methods) on each of these behavioral tests (*Figure 11—figure supplement 2A–C*).

After this initial assessment, we next subjected mice to more challenging balance tests including walking along a narrow balance beam and swimming. On the balance beam, both *Gpr156^del/del^* and *Gpr156^del/+^* mice showed normal coordination (*Figure 11A*). However, when *Gpr156^del/del^* mice were placed in water, they demonstrated severely impaired swimming compared to *Gpr156^del/+^* controls. This impairment was characterized by underwater tumbling and the inability to maintain normal upright posture in the water. All mice were rescued within 60 s and the average time of rescue was significantly lower in *Gpr156^del/del^* compared to control mice (*Figure 11B*; *Gpr156^del/+^* controls

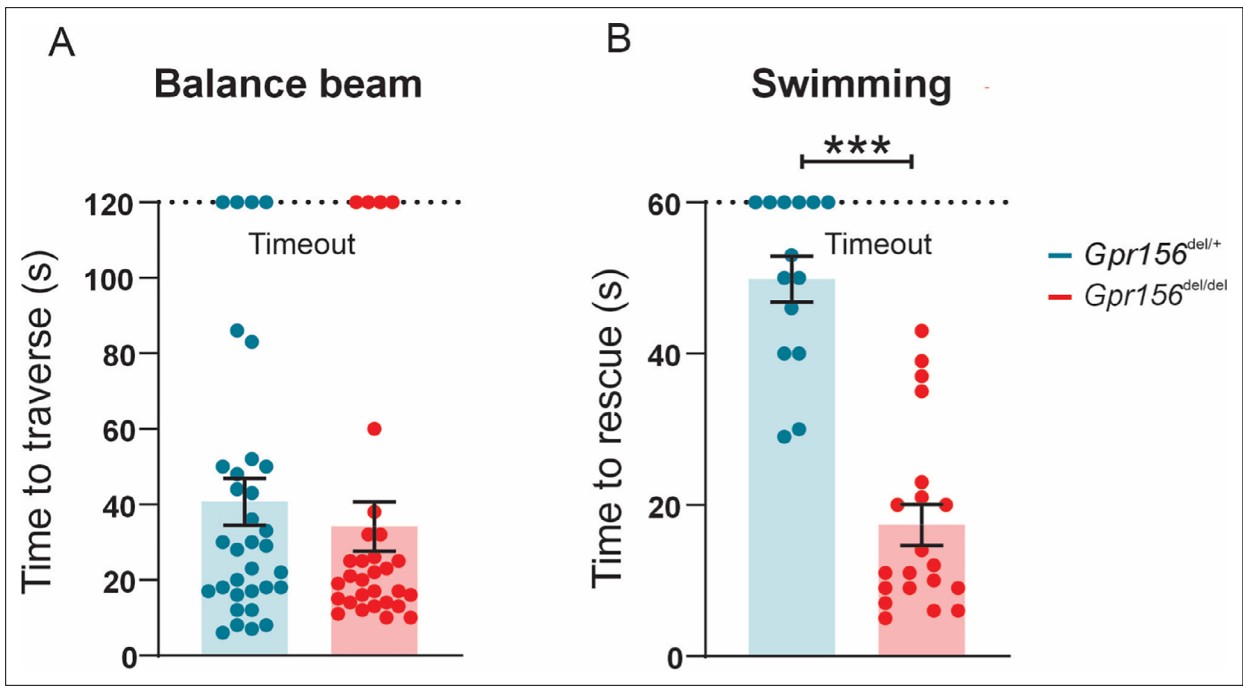

**Figure 11.** *Gpr156* mutant mice failed to swim and maintain their balance in water. (**A**) Time taken to traverse the balance beam for control and *Gpr156^del/del^* mice. N is 33 and 30 for control (green) and *Gpr156^del/del^* (red) mice respectively. (**B**) Time taken to rescue an animal immediately after the animal exhibited underwater tumbling. N is 14 and 20 for control (green) and *Gpr156^del/del^* (red) mice respectively. ***p<0.001.

The online version of this article includes the following figure supplement(s) for figure 11:

**Figure supplement 1.** No resting head tremor was observed in *Gpr156^del/del^* mice.

**Figure supplement 2.** *Gpr156^del/del^* mice scored normally on several basic standard behavioral tests.

= 49.9±3.0s; *Gpr156^{del/del}* = 17.4±2.7s; p<0.0001). Overall, these results indicated that GPR156 is required for vestibular function important for swimming behavior.

We hypothesized that markedly impaired swimming performance could be due to deficits in the otolith rather than semicircular canal dependent pathways. To test this proposal, we quantified eye movements generated by two different vestibulo-ocular reflex behaviors, one driven by activation of the semicircular canals (angular vestibular-ocular reflex, or angular VOR) and the other driven by activation of the otoliths (off-axis vertical rotation, or OVAR).

## Normal aVOR and OKR responses in *Gpr156* mutants

We first quantified the angular vestibulo-ocular reflex in *Gpr156^{del/del}* and *Gpr156^{del/+}* control mice in darkness (VORd). The angular VOR was evoked by rotating the animal sinusoidally around earth vertical axis. To quantify the angular VORd we computed the gain and phase for each testing frequency (0.2, 0.4, 0.8, 1, and 2 Hz,±16°/s). During VORd, both control and *Gpr156^{del/del}* mice displayed robust compensatory eye movements, which increased as a function of frequency (*Figure 12A–B*). Further, to confirm that there was no visual deficit in *Gpr156^{del/del}* mice, we also tested their optokinetic reflex (OKR) by rotating a patterned visual surround sinusoidally about earth vertical axis at the same testing frequencies, while the animal remained stationary (see Methods). The OKR responses of *Gpr156^{del/del}* and control mice were comparable, confirming that there was no visual deficit in *Gpr156* mutants (*Figure 12C–D*). For the sake of completeness, we quantified the angular VOR evoked in the light (VORl) by rotating the animal sinusoidally around earth vertical axis, while the patterned visual surround remained stationary. Again, we found no difference in the VORl response of *Gpr156^{del/del}* versus control mice in this lit environment (*Figure 12E–F*). Finally, we tested whether *Gpr156^{del/del}* mice demonstrated normal ability to adapt their angular VOR. We used a standard angular VOR motor learning protocol (see Methods) in which we carried out 30-minute-long VOR gain-down training for both control and *Gpr156^{del/del}* mice. Head restrained mice were rotated with the visual surround in phase. The training stimulus was 2 Hz with a peak velocity of 16°/s. The learning efficacy was assessed by quantifying change in the VOR gain after training. In both control and *Gpr156^{del/del}* mice, this VOR motor learning protocol produced a significant reduction in VOR gain after training (*Figure 12—figure supplement 1A–B*). Overall, *Gpr156^{del/del}* and control mice demonstrated a comparable percent change of gain decrease at each testing frequencies (*Figure 12—figure supplement 1C*). Together, our VOR and OKR results indicated normal semicircular canals-driven responses in *Gpr156^{del/del}* mice.

## Altered OVAR responses in *Gpr156* mutants

As no defects were apparent in semicircular canal-mediated behavior, we next tested whether otolith-mediated vestibulo-ocular reflex behaviors were impaired in *Gpr156^{del/del}* mice. For this assessment, we recorded eye movements while the mouse was tilted 17° off-the vertical axis (i.e. off-vertical axis rotation (OVAR)), and then rotated at a constant velocity (50°/s for 72 s). The eye velocity evoked by this paradigm comprised two different responses: (1) a transient canal-mediated response that decayed over the first 10–15 s and (2) an otolith-mediated steady-state response in which eye velocity oscillated around a constant bias with a sinusoidal waveform (*Figure 12G*). Overall, our analysis revealed that transient OVAR responses did not differ between control and *Gpr156^{del/del}* mice, supporting the proposal that absence of GPR156 did not affect canal-related ocular responses. In contrast, steady-state OVAR responses demonstrated a significantly impaired bias in *Gpr156^{del/del}* mice as compared to controls (*Figure 12G–H*). The OVAR of *Gpr156^{del/del}* mice was characterized by a significantly reduced bias relative to controls (*Gpr156^{del/del}*: bias = 0.039 ± 0.26 versus control: bias = 3.85 ± 0.54, p=4.03E-5). Thus taken together, these results show that the absence of HC reversal in *Gpr156^{del/del}* alters otolith-mediated OVAR responses. This in turn may explain the swimming deficits observed (*Figure 11B*).

## Discussion

### Hair bundle orientation vs. zonal identity

Taken together, our observations in utricles lacking GPR156 agree with other evidence that in mouse vestibular epithelia, HC orientation is controlled separately from other zonally differentiated properties of either HCs or afferents. Prior evidence includes data from *Emx2* knockouts also exhibiting

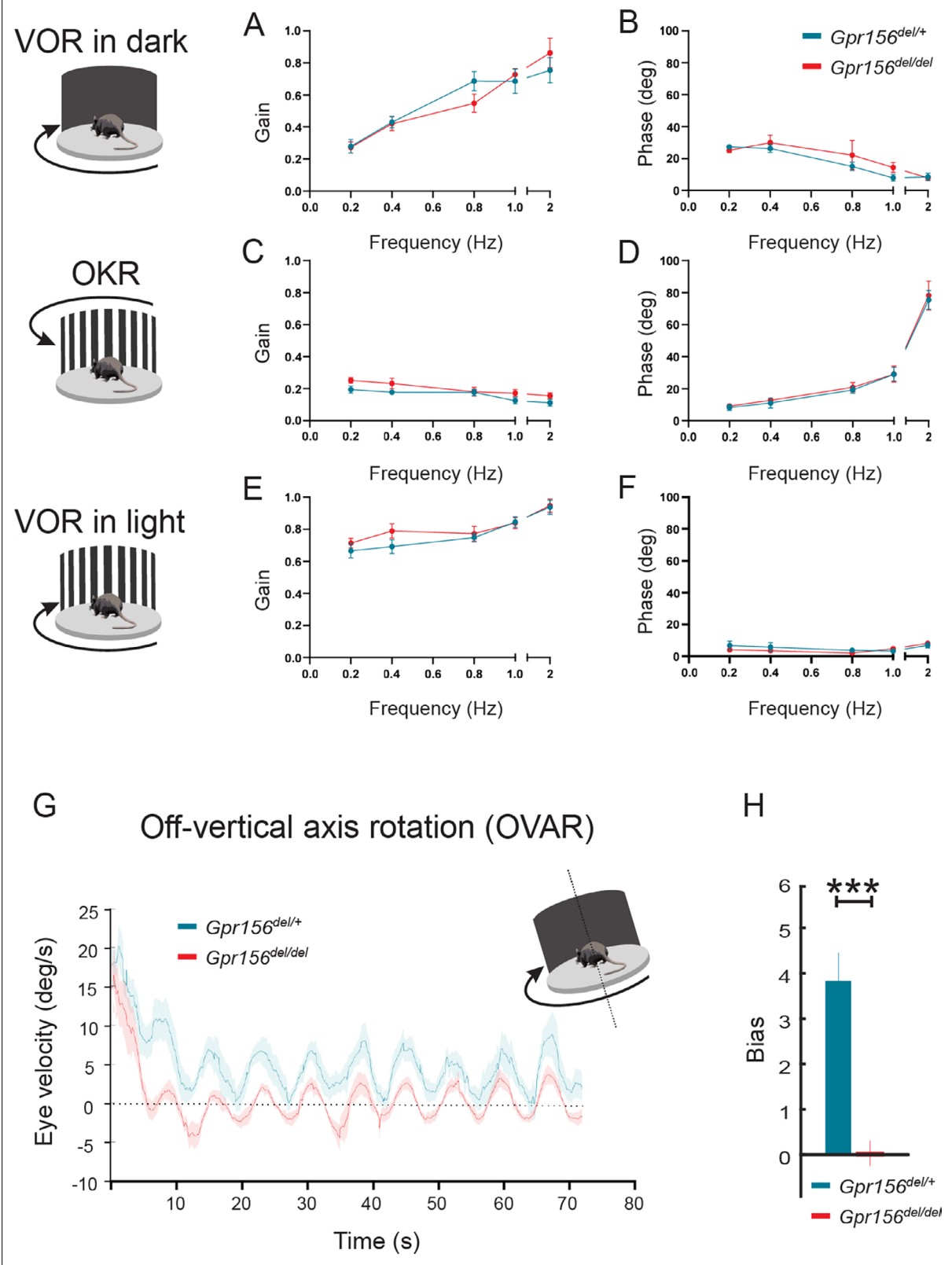

**Figure 12.** VOR and OKR responses were normal but OVAR responses were altered in *Gpr156^{del/del}*mice. (**A–B**) VORd gain and phase (mean ± SEM) plotted as a function of frequency for control and *Gpr156^{del/del}* mice. N is 8 and 7 for control (green) and *Gpr156^{del/del}* KO (red) mice respectively. (**C–D**) OKR gain and phase (mean ± SEM) plotted as a function of frequency for control and *Gpr156^{del/del}* mice. (**E–F**) VORl gain and phase (mean ± SEM) plotted as a function of frequency for control and *Gpr156^{del/del}* mice. (**G**) Average eye velocities (mean ± SEM) during 72s-long off-vertical axis rotation

*Figure 12 continued on next page*

*Figure 12 continued*

for control and *Gpr156^del/del* mice. N is 8 and 7 for control (green) and *Gpr156^-/-* (red) mice respectively. (**H**) OVAR bias during the steady state (mean ± SEM) for control and *Gpr156^del/del* mice. \*\*\*p<0.001.

The online version of this article includes the following figure supplement(s) for figure 12:

**Figure supplement 1.** VOR learning was unimpaired in *Gpr156^del/del* mice.

loss of the LPR without altering zoning (*Jiang et al., 2017*) and data showing that inactivating cytochrome P450 family 26 deletes striolar identity without affecting HC orientation (*Ono et al., 2020*). This conclusion stands in contrast to zebrafish lateral line results, where transduction properties vary with HC orientation depending on Emx2 (*Kindig et al., 2023*) and Gpr156 (this study).

## Asymmetrical mechano-electrical transduction

Zebrafish lateral-line HCs use a combination of 2 mechanosensitive channels, Tmc2a and Tmc2b (*Chou et al., 2017*). In the posterior lateral line, Emx2$^+$ HCs that sense posterior flow rely on primarily on Tmc2b, while Emx2$^-$ HCs that sense anterior flow rely on both Tmc2a and Tmc2b. Recently, a functional asymmetry was discovered between anterior- and posterior-sensitive HCs: Emx2$^+$ HCs that rely primarily on Tmc2b have smaller mechanosensitive responses compared to Emx2$^-$ HCs that rely on both Tmc2a and Tmc2b (*Kindig et al., 2023*). This work raises the possibility that the transcription factor Emx2 inhibits *tmc2a* transcription to reduce mechanosensitive responses in HCs sensing posterior flow. In our present study we find that functional asymmetry was also lost in *gpr156* mutants (*Figure 5*). In the context of mouse HC orientation, EMX2 is necessary and sufficient to enrich and polarize GPR156 at HC apical junctions and reverse HC orientation (*Kindt et al., 2021*). As the Gpr156 receptor is unlikely to act transcriptionally, zebrafish Emx2 may act through Gpr156 to impact Tmc2a/2b function. For example, polarization of Gpr156 at apical junctions may limit scaffolding or transport of Tmc proteins to homomeric channels composed of Tmc2b. Overall, our work in zebrafish indicates that both Emx2 and Gpr156 are important to dictate HC orientation and the mechanosensitive properties of HCs with opposing orientations.

We did not see a clear impact of GPR156 loss on the transduction properties of *Emx2$^+$* HCs in the mouse utricle (*Figures 2–4*). Subtle effects remain possible, however, given the variance in single-cell electrophysiological data from both control and mutant mice. Nevertheless, current results are consistent with normal HC function in the *Gpr156* mouse mutant, a prerequisite to interrogate how non-reversed HCs affects vestibular behavior.

## Importance of the LPR for vestibular function

The brain combines information from the maculae and cristae with inputs from other modalities, including the somatosensory and visual systems, to compute a representation of our self-motion and spatial orientation (reviewed in *Cullen, 2019*). Our present findings demonstrate that GPR156 is *not* required for macular HCs to selectively contact different afferents across the LPR. Accordingly, we predicted that HCs that fail to reverse in the lateral utricle of *Gpr156* mutants would lead to deficits in paradigms that test macular contributions to behavioral responses. Indeed, *Gpr156* mutant mice demonstrated impaired performance on two tests that engage the otolith organs: swimming and off-vertical-axis rotation (OVAR). First, during swimming, impaired spatial orientation results from the combined altered otolith input and reduced somatosensory feedback due to the aqueous environment. Second, *Gpr156* mutant mice displayed impaired sustained responses to off-vertical axis rotation (OVAR). While initial eye movement response to OVAR is mediated by stimulation of semicircular canal HCs, the canal response decays such that sustained eye movement is solely driven by the otolith stimulation (*Beraneck et al., 2012*; *Ono et al., 2020*). Quantification of the sustained responses revealed a marked reduction in this otolith-mediated reflex response. Confirming that these deficits are specific to otolith function, the *Gpr156* mutant mice did not show overt vestibular dysfunction such as spinning or head tilt that can arise with global dysfunction of vestibular epithelia. Further, they displayed normal performance during testing that selectively engaged superior canal organs, namely angular VOR in the light and dark. This latter finding was predicted; because HC orientation is unchanged in *Gpr156* mutant cristae, the coding properties of semicircular canal afferents should

be unchanged and would therefore produce a normal angular VOR upon specific stimulation of canal ampullae.

## Comparing vestibular phenotypes of *Gpr156* and *Emx2* mutants

*Gpr156* does not appear to be significantly expressed in the central nervous system according to the Allen Brain Atlas (*Lein et al., 2007*). In addition, *Advillin-Cre* tracing and dye-filling experiments (*Figures 6, 7*) showed that vestibular ganglion neurons still selectively innervated LES HCs even when lacking GPR156. We thus speculate that HC signals in the LES zone of *Gpr156* mutants are relayed to the cerebellum (*Ji et al., 2022*; *Maklad et al., 2010*) via the central projection of *Advillin*+ neurons as usual, but with abnormal (non-reversed) response polarity. Overall, it appears unlikely that vestibular deficits observed in our constitutive *Gpr156* mouse mutants reflect defective neuronal activity in addition to a missing LPR. That said, we cannot entirely rule out another role of GPR156 beside its regulation of HC orientation. This could be addressed in a follow-up study by limiting *Gpr156* inactivation to HCs.

It is interesting to compare vestibular-related behaviors between the *Gpr156* and *Emx2* mouse models in light of established similarities and differences in protein distribution and function (*Ji et al., 2022*; *Jiang et al., 2017*; *Kindt et al., 2021*). Regarding behavior, this study and work by *Ji et al., 2022* agree that vestibular deficits in each model are relatively mild (normal balance beam, VOR and OKR). *Emx2* inactivation in HCs (*Gfi1-Cre*) showed impaired swimming ability, but defects were less severe and different in nature compared to constitutive *Gpr156* mutants. *Gpr156* mutants displayed substantial difficulty in maintaining their orientation in water, and frequently rolled over such that they needed to be rescued. In contrast, *Emx2* mutants did not need rescue and were described as demonstrating 'frantic' swimming behavior, spending more time trying to climb out of the water as compared to their littermate controls. OVAR highlighted otolith-specific defects in *Gpr156* mutants but was not performed in *Emx2* mutants.

GPR156 acts downstream of EMX2 to regulate hair bundle orientation, and perhaps mechanosensitivity in zebrafish. Another significant new result in the current study is to show that afferent projections segregate with HC orientation and not with zonal identity (*Figure 7*; Advillin neuron labeling), yet segregation does not depend on HC orientation since it is unchanged in mouse and zebrafish *Gpr156* mutants (*Figures 6–10*). It follows that EMX2 must influence afferent segregation (*Ji et al., 2022*; *Ji et al., 2018*) via other effectors unrelated to HC orientation. Accordingly, Ji and colleagues concluded that EMX2 regulates HC orientation and afferent patterning independently, with *Emx2* expression in support cells critical for proper afferent contacts with HCs (*Ji et al., 2022*). This aligns well with previous evidence that *Gpr156* is only expressed in HCs (*Kindt et al., 2021*).

Ji and colleagues also used specific inactivation of the transduction protein TMIE in the LES (*Emx2-Cre*) to silence the *Emx2*+ HC population. Interestingly, as with *Emx2* mutants, they did not observe severe behavioral defects, and swimming ability in particular was not obviously compromised in contrast to *Gpr156* mutants. It should be noted however that while the *Gpr156* KO and *Emx2-Cre; Tmie* cKO models both lack a normal complement of bidirectional responses, they provide different forms of aberrant linear acceleration signals. In the *Tmie* cKO model, LES HC inputs to the cerebellum are missing but in the *Gpr156* model they are retained and of opposite nature since LES HCs fail to reverse their orientation.

Several considerations could potentially explain why the *Gpr156* model appears to display more severe behavioral defects compared to the *Emx2* models. First, HC or afferent properties were not tested in *Emx2* mutants (*Ji et al., 2022*) and may be defective. Second, compounded HC orientation and afferent defects in the *Emx2* mutants could dampen the deleterious consequence of having LES HCs drive a neuronal response of reversed polarity. Afferent defects in *Emx2* mutants were reported during embryogenesis (E16.5) and innervation could further degenerate in adults, further dampening the consequence of non-reversed HC activity in the LES. It is worth noting that aberrant vestibular afferent signals are clinically more deleterious than absent signals, which perhaps explains why properly relayed yet aberrant signals from LES HCs in *Gpr156* mutants result in behavioral defects worse than observed in *Emx2* or *Emx2-Cre; Tmie* cKO models.

In conclusion, the *Gpr156* mutant mouse model is well-suited to specifically interrogate how HC orientation reversal and the resulting LPR influence vestibular function because it appears to lack confounding afferent patterning defects. In future studies, this model could be improved by limiting *Gpr156* inactivation to HCs.

## Methods

### Mouse strains and husbandry

The *Gpr156*<sup>del</sup> strain (*C56BL/6N-Gpr156tm1.1(KOMP)Vlcg/J*; MGI:5608696) was produced by the Knockout Mouse Project (KOMP) and studied previously in *Kindt et al., 2021*. The *Advillin-Cre* strain is *B6.129P2-Avil*<sup>tm2(cre)Fawa/J</sup> (MGI: 6196038) (*Zhou et al., 2010*). The *Tigre Ai140* strain is *B6. Cg-Igs7*<sup>tm140.1(tetO-EGFP,CAG-tTA2)Hze</sup> (MGI: 5904001; *Daigle et al., 2018*). Some whole-cell patch clamp recordings were obtained from CD1 mice (Charles River Laboratories USA). All animals were maintained in standard housing and all animal work was reviewed for compliance and approved by the Animal Care and Use Committees of The Jackson Laboratory (anatomical experiments), the University of Chicago (cellular electrophysiology and afferent labeling experiments), and the Johns Hopkins University School of Medicine (behavioral experiments).

### Mouse immunofluorescence and imaging (Figure 1, Figure 7)

Temporal bones of adult animals were isolated, the oval and round windows were cleared and the cochlear apex was punctured to allow better access to the fixative. Samples were then fixed in paraformaldehyde (PFA 4%) overnight at 4 °C, rinsed in PBS and incubated overnight in 4% Ethylenediaminetetraacetic acid (EDTA) at room temperature for decalcification. Both the "trio" (utricle, anterior crista, horizontal crista) and the saccule were dissected from the bone and their epithelium exposed. After dissection, samples were permeabilized and blocked in PBS with 0.5% Triton-X100 and bovine serum albumin (1%) for at least at 1 hr at room temperature. For SPP1/Osteonpontin staining, temporal bones were acutely dissected to isolate the trio and the saccule and expose their epithelium, and an eyelash was used to scrape off the otoconia. Samples were then fixed in PFA 4% for 1 hr at room temperature before being permeabilized and blocked as described previously. Primary and secondary antibodies were incubated overnight at 4 °C in PBS, and fluorescent dye-conjugated phalloidin was added to secondary antibodies. Samples were washed three times in PBS +0.05% Triton-X100 after each antibody incubation before being finally post fixed in PFA 4% for at least 1 hr at room temperature. Primary antibodies used were:

> Mouse anti-MYO7A (Developmental Studies Hybridoma Bank/DSHB 138–1, 1:1000)
> Goat anti-OCM (ThermoFisher Scientific, PA547832, 1:200)
> Goat anti-SPP1 (R&D Systems, AF808, 1:200)
> Rabbit anti-PCNT (Biolegend, PRB432C, 1:400)
> Goat anti-SOX2 (R&D Systems, AF2018, 1:200)

Secondary antibodies were raised in donkey and coupled with AlexaFluor (AF) 488, 555, or 647 (donkey anti-rabbit 488 (A-21206), donkey anti-mouse 647 (A-31571), donkey anti-goat 555 (A-21432); ThermoFisher Scientific). Fluorescent conjugated phalloidin was used to reveal F-actin (CF405 (00034); Biotum).

Confocal images were captured on a line scanning confocal microscope (LSM800) using Zen 2.6 software, the Airyscan detector in confocal mode, and either a 20× or a 63× 1.4 NA oil objective lens (Carl Zeiss AG). Images show a single optical plane unless stated otherwise in the figure legend.

### Mouse immunostain quantification (Figures 1 and 7)

All images were processed using Adobe Photoshop (CC2020), and the same image treatment was applied across genotypes for each experiment. All quantifications included at least three animals of each genotype, and all values plotted in the study, as well as animal cohort size (N), HC number and stereocilia number (n) are detailed in legends. To quantify striolar surface, striolar density, as well as HC Type I and II density by region, images were captured using a DM5500B fluorescence microscope using the Leica Application Suite (LASX) and a 20 x objective (Leica). Striolar surface was measured using the polygon selection and area tools in Fiji/ImageJ based on the Oncomodulin staining. Striolar density was measured by using the cell counter tool in Fiji/ImageJ to obtain the total number of Oconmodulin positive HCs. That number was then normalized by the total striolar surface previously measured to obtain a final number of HCs per 1000 $\mu m^2$. To measure Type I and Type II HC density by region, three regions of interest (ROI) were defined centrally of either 130x50 $\mu$m (utricle) or 150x40 $\mu$m (saccule). The lateral extrastriolar (LES) domain was placed 20 $\mu$m inside the macula starting from the lateral edge of the utricle, encompassing most of the region lateral to the line of

polarity reversal (LPR). The striolar domain was defined immediately adjacent to the LES domain based on the LPR and oncomodulin signal in controls. The medial extrastriolar (MES) domain started 50 µm medial to the striolar domain. The anterior extrastriolar domain (ANT) was placed 10 µm from the anterior edge of the saccule. The striolar domain was defined 20 µm posterior to the ANT domain based on oncomodulin staining in controls, and the posterior extrastriolar domain (POST) was defined 50 µm posterior to the striolar domain. The location of these domains is illustrated in *Figure 1A*. The same fields and spacing were next used for the *Gpr156^{del/del}* mutants. The cell counter plugin in Fiji/ImageJ was used to count HCs in each ROI.

## Data analysis and statistics – mouse anatomy (Figures 1 and 7)

All data were plotted in Prism 9 (GraphPad). Striolar surface, HC density and number of Type I and type II HCs in each region of the maculae were plotted individually. Their distribution was framed by 25–75% whisker boxes where exterior lines show the minimum and the maximum, the middle line represents the median, and + represents the mean. Statistical significance was tested using Mann-Whitney U (non-parametric, unpaired t-test) for striolar surface and HC density, and 2-way ANOVA with Sidak's multiple comparison for type I and type II HCs.

## Mouse hair cell and afferent electrophysiology (Figures 2, 4 and 8)

We followed procedures described in *González-Garrido et al., 2021* to prepare and record from HCs and calyceal afferent terminals in semi-intact utricles comprising the sensory epithelium and attached distal vestibular nerve including the vestibular ganglion. Recordings were analyzed for 83 cells from 57 *Gpr156* heterozygotes and 93 cells from 73 *Gpr156*-null animals.

### Preparation

Following protocols approved by the University of Chicago Animal Care and Use Committee, mice were deeply anesthetized with gaseous isoflurane and then decapitated to allow dissection of the utricles and attached nerve and ganglion. The preparations were dissected in our *standard bath (external) solution*: cold Leibovitz's-L15 medium (L15, Gibco, #41300–021) supplemented with 10 mM HEPES (4-(2-hydroxyethyl)–1-piperazineethanesulfonic acid; ~315 mmol/kg, pH 7.4). The utricles were then treated with L15 containing proteinase XXIV (100 mg/ml, Sigma, St. Louis, MO #P8038) for 11 min at room temperature to facilitate mechanical removal of the otolithic membrane. The epithelium was mounted on a cover slip with glued glass fibers and placed in a recording chamber on a microscope (Zeiss, Axio Examiner A1) equipped with Nomarski and fluorescence optics.

### Recording

For whole-cell recording, pipettes (R6, King Precision Glass) were heat-pulled (PC100, Narishige) and parafilm-wrapped to reduce capacitance. Pipette resistances were 4–5 MΩ for the standard internal and external solutions. Standard internal solution contained (in mM): 135 KCl, 0.1 CaCl2, 3.5 MgCl2, 3 Na2ATP, 5 creatine phosphate (Na salt), 0.1 Na-cAMP, 0.1 Li-GTP, 5 EGTA, and 10 HEPES, plus ~28 mM KOH to bring pH to 7.3 and osmolality to ~300 mmol/kg. Internal solution was supplemented with sulforhodamine 101 (1 mg/100 ml; Thermo Fisher Scientific) for visualization of recorded hair cells. The bath (external solution) was modified L15 (described above), which was perfused during the experiment.

We were able to locate LES, striola or MES despite loss of the LPR because anatomical features of the striola are maintained in the otolith organs of *Gpr156^{del/del}* mice (*Kindt et al., 2021*), and some of these (hair bundle size, spacing of HCs, abundance of complex calyces) are visible in live tissue with Nomarski optics. Usually we aimed for HCs or calyces >50 µm away from zonal boundaries to avoid confusion over zone. In one experimental series, however, we focused on the zonal boundary to see whether afferents respected zonal boundaries in Gpr156-deleted mice.

To record in whole-cell mode from a specific HC or afferent calyx, we cleaned the cell membrane by the outflow of pipette solution and applied gentle suction to promote giga-seal formation and membrane rupture. Stimulus protocols and data acquisition were implemented by the patch clamp amplifier (model EPC-10, HEKA Elektronik) controlled by Patchmaster software. Data were digitized at sampling intervals of 10 or 20 microseconds. Pipette and membrane capacitive currents were nulled and series resistance (R_s) was compensated by 80% on-line with the amplifier's controls. Voltages were

corrected offline for the liquid junction potential (–4 mV) and the voltage error due to residual $R_s$ (20% of total $R_s$, 8.1±0.1, n=207).

## Mechanical stimulation of mouse hair cells

Hair bundles were deflected with a rigid probe (pulled glass pipette, BF120-60-10, Sutter instrument Company) glued to a piezoelectric bimorph ceramic and brought into contact with the staircase of stereocilia ('back' of the bundle) at ~half-height. Displacement of the probe to driving voltage was calibrated with a CCD camera, yielding a scale factor 330–410 nm/V, depending on the probe. Probe motion recorded with a photodiode (PIN-6D, United Detector Technology) revealed a primary resonance at 1900–2000 Hz. To attenuate this ringing, the voltage input to the bimorph was low-pass filtered (8-pole Bessel filter, Model 900, Frequency Devices) set at 1 kHz, for a step rise time (10–90%) of ~300 µs.

Mechanotransduction current was recorded in voltage-clamp mode with the holding potential (HP) of –94 mV and –84 mV for type I and type II HCs, respectively. The larger negative potential for type I HCs was chosen to reduce $g_{K,L}$ and so improve the voltage clamp. Responses to three identical displacement step protocols were averaged and low-pass filtered off line (corner frequency, 2 kHz, 8-pole Bessel filter, as above). Step protocols comprised either 20 or 40 displacement steps iterated by 100 nm or 35 nm, respectively, and beginning slightly negative to the resting bundle position. Transduction sensitivity and adaptation was analyzed in 49 HCs with $I_{max}$ >150 pA (23 HCs from heterozygotes and 26 HCs from null mutants), as described next.

## Quantification of mouse electrophysiological data

Analysis of $I_{MET}$ properties is described elsewhere (*Songer and Eatock, 2013*; *Vollrath and Eatock, 2003*). Briefly, conductance-displacement [G(X)] activation curves were calculated by dividing $I_{MET}$ (averaged across 2–4 presentations) by the driving force (HP – $I_{MET}$ reversal potential, 0.2 mV; *Corns et al., 2014*) and plotting $G_{MET}$ against probe displacement (*X*). Probe displacement as a function of driving voltage was calibrated off-line with a photodiode (*Songer and Eatock, 2013*). G(X) curves were averaged across HCs and fitted with a Boltzmann function (*Equation 1*):

$$G_{MET}(X) = \frac{G_{max} - G_{min}}{1 + e^{\frac{(X_{1/2} - X)}{S}}} + G_{min} \tag{1}$$

where $G_{max}$ and $G_{min}$ are maximum and minimum MET conductance, respectively, $X_{1/2}$ is displacement that evokes half-maximal $G_{MET}$ and $S$ is displacement corresponding to an e-fold rise in $G_{MET}$. We used the Boltzmann fits to calculate operating range (OR): displacement range corresponding to 10–90% of the response range, $G_{max} - G_{min}$.

We characterized transducer adaptation for responses at $X_{1/2}$, shown previously to reveal both fast and slow components of adaptation in mouse utricular hair cells (*Vollrath and Eatock, 2003*). To reduce noise, current responses were averaged (*n* usually 3) and low-pass filtered at 1.5 kHz. Fast and slow adaptation components were estimated from double-exponential fits of the first 300ms of the step response (*Equation 2*). Deviation of those fits at the onset response signaled a very fast transducer adaptation component, which was separately estimated with a single-exponential fit of the first 1.5 ms of the response (*Equation 3*).

$$I(t) = A_f e^{\frac{-t}{\tau f}} + A_s e^{\frac{-t}{\tau s}} + I_{SS} \tag{2}$$

$$I(t) = A_{vf} e^{\frac{-t}{\tau vf}} + I_{SS} \tag{3}$$

Here, $\tau_{vf}$, $\tau_f$ and $\tau_s$ are the very short (1ms or less), short (from 1 to 10ms) and long (>10ms) time constants, respectively; $A_{vf}$, $A_f$ and $A_s$ are the amplitudes of the corresponding exponential terms; and $I_{SS}$ is the steady-state current. All HCs had a very fast or fast component or both; some lacked a slow component that was detectable in the 400 ms step.

We calculated extent of adaptation (% decay) at $X_{1/2}$ as follows (*Equation 4*):

$$\text{Extent of adaption} = \left(\frac{I_{peak} - I_{SS}}{I_{peak} - I_0}\right) \times 100 \tag{4}$$

where $I_0$, $I_{peak}$, and $I_{SS}$ are pre-step, steady-state, and peak currents, respectively.

In mouse utricular HCs, the dominant voltage-gated currents are carried by outwardly rectifying $K_V$ channels (*Martin et al., 2024*). By the end of the first postnatal week, these currents provide distinctive electrophysiological signatures in type I and type II mouse utricular HCs, which include more negatively activating $K^+$ currents and more negative resting potentials for type I HCs (*Rüsch and Eatock, 1996*; *Rüsch et al., 1998*). To characterize the voltage dependence of the currents, we applied families of iterated test voltage steps (400ms duration) from holding potentials of –74 mV (type I HCs) and –64 mV (type II HCs). The different holding potentials reflect the difference in resting potentials of the two HC types. Steady-state voltage dependence was measured by constructing conductance-voltage [G(V)] relations from tail currents, measured 1ms after the 400 ms test voltage steps, at a common membrane voltage of –39 mV (chosen to reduce contamination of tail currents by HCN currents). Tail current was converted to 'tail conductance' by dividing by driving force ($V_m - V_{rev}$), where $V_{rev}$ is approximated by the $K^+$ equilibrium potential (–84 mV in our solutions). Tail conductance was plotted against the iterated test step voltage and fit with the Boltzmann function:

$$G(V) = \frac{G_{max} - G_{min}}{1 + e^{(V_{1/2} - V)/S} + G_{min}} \tag{5}$$

where $V_{1/2}$ is the voltage of half-maximal activation and $S$ is the voltage change over which G changes e-fold at voltages negative to $V_{1/2}$. Voltage was corrected for voltage errors caused by residual $R_S$ and for the liquid junction potential by subtracting 4 mV. Input resistance ($R_{in}$) was estimated from current clamp data as the slope of a linear regression of the $V_m$(I) relation for small currents around resting potential.

## Imaging of afferent terminal arbors in mouse utricle (Figure 6)

To investigate whether afferent receptive fields (terminal arbors) were disrupted by the lack of bundle reversal in the lateral extrastriola (LES), we recorded in whole-cell mode from calyces near the LES-striolar border. To label the terminal arbors for imaging, we included Alexa Fluor 594 (#A10438 Thermo Fisher Scientific) instead of sulforhodamine in the pipette solution (50 micromolar). After whole-cell recording was achieved, we waited 20 min as the fluorescent pipette solution diffused into the calyx and throughout the terminal arbor, then removed the pipette and fixed the utricular epithelium with 4% paraformaldehyde for 10 min at room temperature (~22°). Fixed utricles were immersed in blocking buffer (PBS with 4% normal donkey serum and 0.2% triton) then treated with primary antibodies diluted with blocking buffer overnight at 4 °C. Primary antibodies were:mouse monoclonal anti-βII spectrin (1:500; BD Transduction, #612562), to label hair bundles and show the location of the kinocilium by absence of label; rabbit polyclonal anti-calbindin (1:250; Invitrogen, #711443) to label the striola and juxtastriola (*Leonard and Kevetter, 2002*). Specimens were then washed extensively and incubated with secondary antibodies conjugated with fluorescent proteins (donkey anti-mouse or rabbit IgG (H + L) antibody, ThermoFisher Scientific) for 1 hr at 4 °C. Stained specimens were mounted whole in mounting medium (Vector Laboratories, #H-1500), imaged with spinning-disc confocal microscopy (Marianas SDC 3i), and analyzed with Adobe Photoshop 2021 and FluoRender (ver. 2.26.3).

## Measurement of excitability from mouse utricular afferents (Figure 8)

To document excitability of afferents in the LES, we recorded from calyces in whole-cell current clamp mode, injected current steps and noted the thresholds for spiking and the firing patterns. Most LES afferents are dimorphic, forming both calyces and boutons on type I and II HCs, respectively. We rejected calyces with resting potentials less negative than –50 mV. We evoked spikes with current steps to determine if afferents had step-evoked firing patterns consistent with expected zonal patterns of spiking regularity in mammalian vestibular epithelia (*Goldberg et al., 1990*; *Iwasaki et al., 2008*; *Kalluri et al., 2010*). Families of 500 ms current steps were applied from zero-current potential, beginning at –200 pA and incremented by 50 pA. Threshold current ($I_{thresh}$) was taken as the minimum current (±50 pA) required to evoke any spikes.

We classified neuronal firing patterns according to the number of spikes evoked by a 400 ms step at $I_m = 3 \times I_{thresh}$ (**Ono et al., 2020**): a pattern of 1–2 spikes at step onset was considered 'transient'; if more spikes were evoked, the pattern was 'sustained'. These patterns have been associated with the mechanisms that give rise to well-known differences in the regularity of firing across zones. The transient step-evoked pattern is associated with striolar, irregular afferents and the sustained step-evoked pattern with extrastriolar, regular afferents. The difference in step-evoked firing patterns is attributed to zonal differences in expression of $K_V$ channels that influence spike regularity (**Iwasaki et al., 2008**; **Kalluri et al., 2010**) and, more importantly, encoding strategy of afferent fibers (**Jamali et al., 2019**).

## Data analysis and statistics – mouse electrophysiology (Figures 2, 4 and 8)

Data analysis, statistical tests, curve fitting and graphing were done with OriginPro (2022; OriginLab, Northampton, MA), Clampfit (Molecular Probes) and Matlab (Mathworks, Natick, MA). For samples that had homogeneous variance, significance was tested with either a two-tailed t-test or a one-way ANOVA with post-hoc Tukey test. When samples had unequal variance, we used either Welch's t-test or Kruskal-Wallis ANOVA with post-hoc Mann-Whitney U test. For significant differences ($p<0.05$), we calculated effect size with Hedge's g equation. For non-significant (NS) differences, we calculated statistical power post hoc (OriginPro 2022) and these values are entered in tables (**Supplementary files 1–5**) along with p-values.

## Quantification of vestibular reflexes in mice (Figures 11 and 12)

We tested the angular vestibulo-ocular reflex (aVOR), the optokinetic reflex (OKR), and the otolith vestibulo-ocular reflex (off-vertical axis rotation, OVAR). Specifically, aVOR tests semicircular canal function, whereas OVAR tests otolith function. For completeness we also tested each animal's OKR as a control of the integrity of visuo-motor function.

### aVOR and OKR

Surgical techniques and experimental setup have been previously described (**Beraneck and Cullen, 2007**). Eye movement data were collected using an infrared video system (ETL-200, ISCAN system). The rotational velocity of the turntable (head velocity) was measured using a MEMS sensor (MPU-9250, SparkFun Electronics). Eye movements during the OKR were evoked by sinusoidal rotations of a visual surround (vertical black and white stripes, visual angle width of 5°) placed around the turntable at frequencies 0.2, 0.4, 0.8, 1, 2, and 3 Hz with peak velocities of ±16°/s. To record VOR responses, the turntable was rotated at sinusoidal frequencies 0.2, 0.4, 0.8, 1, 2, and 3 Hz with peak velocities of ±16°/s in both light and dark. In both conditions, the visual surround remained stationary, but in the dark condition all lighting was extinguished so that surroundings were made dark. Head and eye movement signals were low-passed filtered at 125 Hz and sampled at 1 kHz. Eye position data were differentiated to obtain velocity traces. Cycles of data with quick phases were excluded from the analysis. Least-square optimization determined the VOR and OKR gains, and phases plotted as mean ± standard error of the mean (SEM) against all frequencies for all mice.

### OVAR

Techniques employed for measurement of eye movements during OVAR in alert mice were described elsewhere (**Beraneck et al., 2012**). Briefly, recordings made after fixating mice on a rotating platform, which was tilted 17° with respect to the ground. The platform speed was increased from 0 to 50°/s in 500ms and maintained its constant velocity for 72 s (10 complete rounds) before being stopped. Eye movements were measured using the same Infrared video system (ETL-200, ISCAN system) used for angular VOR and OKR recordings. Quick phases were identified as previously described and excluded from subsequent analysis. We then estimated the time constant of the OVAR slow-phase eye velocity response decay, as well as the amplitude and phase of its sinusoidal modulation using a linear regression approach.

## Mouse swimming assessment

A large container (26.25×16.25 × 14.38') was filled with water (24–26°C) at the height of at least 15 cm. The mouse was placed into the water and observed for up to 1 min. Its performance was rated using the following scale (*Hardisty-Hughes et al., 2010*):

> 0=swims, body elongated, and tail propels in flagella-like motion.
> 1=immobile floating.
> 2=underwater tumbling.

In a time-to-rescue trial, if an animal exhibited distressed swimming (underwater tumbling) before 60 s, the animal was rescued and time of rescue was marked. If not, the trial was marked as lasting 60 s.

## Mouse balance beam assessment

A 6-mm-wide and 40-cm-long beam was used for balance beam testing. The mice traversed 40 cm to a dark box. Walking speed was measured by recording the time the mouse took to reach the goal box from the opposite end of the beam. Mice were scored 'time out' if they failed to reach the endpoint in 2 min.

## Air righting, tailing hanging, and contact inhibition of righting tests
### Air righting test

The mouse was picked up by the tail and lowered into a container (45cm long x 30 cm wide x 30 cm deep) so that all four feet touched the bottom. The container was quickly inverted at a height of 30–40 cm so mouse fell supine onto a foam cushion below. How the mouse landed was observed and rated as below (*Rabbath et al., 2001*):

> 0=the animal lands on its feet (normal).
> 1=the animal lands on its side (mild deficit).
> 2=the animal lands on its back (severe deficit).

## Tail hanging test

The animal was picked up by the tail and lowered to an even surface, and its posture rated using the following scale (*Rabbath et al., 2001*):

> 0=straight body posture with extension of forelimbs toward the earth (normal).
> 1=slightly bending the body ventrally (intermediate response).
> 2=persistently bending the body (severe response).

## Contact inhibition of righting test

The mouse was picked up by the tail and lowered into a small container so that all four feet were in contact with the bottom, then the top of the container closed in contact with the mouse's back. The container was then quickly inverted so that the mouse became supine while the surface remains in contact with the soles of the mouse's feet. The mouse's reflex was rated using the following scale (*Rabbath et al., 2001*):

> 0=animal rights successfully (normal).
> 1=partial righting (intermediate response).
> 2=complete loss of righting (severe response).

## Quantification of tremor during resting head movement (Figure 11—figure supplement 1)

Mice were placed in a cylinder (9 cm diameter and 21.5 cm height) that limited their motion, so mice maintained their steady posture. Head movements in six dimensions were recorded for 2 min using a miniature head motion sensor (MPU-9250, SparkFun Electronics, Niwot, CO, United States) affixed on the top of the skull, which comprises a three-dimensional (3D) accelerometer (measures linear

acceleration; fore/aft, lateral, and vertical) and 3D gyroscope (measures angular velocity: roll, pitch, and yaw). Data was acquired at 200 Hz using windows-based CoolTerm software. We then computed the power spectral densities (pwelch function, MATLAB, MathWorks) using Welch's averaged periodogram with nfft = 4096 and a Bartlett window (4096ms duration) for all six dimensions of movement.

## Data analysis and statistics – mouse behavior (Figures 11 and 12)

Data are reported as the mean ± SEM. Nonparametric Mann-Whitney U-test was performed to test significance for time to traverse (balance beam) and time to rescue (swimming) between two groups. Two-way repeated-measures ANOVA followed by Bonferroni post hoc comparison tests was used for the VOR data across frequencies. For power spectra analysis we used an independent sample permutation test to test significant differences between the two groups. Prism 9 (GraphPad) or MATLAB was used for statistical analyses. The same independent sample permutation test was also used to test significance for bias and gain of OVAR responses.

## Zebrafish strains and husbandry

Zebrafish (*Danio rerio*) were grown at 30 °C using a 14 hr light, 10 hr dark cycle. Larvae were raised in E3 embryo medium (5 mM NaCl, 0.17 mM KCl, 0.33 mM $CaCl_2$, and 0.33 mM $MgSO_4$, pH 7.2). Larvae were examined at 5–6 days post fertilization (dpf). The following previously established mutant and transgenic zebrafish strains were used in this study: *gpr156$^{idc15}$*, *emx2$^{idc5}$* (*Jiang et al., 2017*; *Kindt et al., 2021*), *Tgmyo6b:memGCaMP6s$^{idc1}$* (*Kindt et al., 2021*). For comparisons, *emx2* mutants were compared to wild-type or *emx2$^{+/-}$* animals. *Gpr156* mutants were compared to *gpr156$^{+/-}$* animals. Zebrafish work was performed at the National Institute of Health (NIH) and approved by the Animal Use Committee at the NIH under animal study protocol #1362–13.

## Zebrafish immunofluorescence and imaging (Figure 5, Figures 9 and 10)

Immunohistochemistry was performed on whole larvae at 5 dpf. Whole larvae were fixed with 4% paraformaldehyde in PBS at 4 °C for 3.5 hr. For pre- and post-synaptic labeling all wash, block and antibody solutions were prepared with 0.1% Tween in PBS (PBST). For Emx2 labeling performed on sparse afferent labeling (see below) all wash, block and antibody solutions were prepared with PBS +1% DMSO, 0.5% Triton-X100, 0.1% Tween-20 (PBDTT). After fixation, larvae were washed 4×5 min in PBST or PBDTT. For synaptic labeling, larvae were permeabilized with Acetone. For this permeabilization larvae were washed for 5 min with $H_2O$. The $H_2O$ was removed and replaced with ice-cold acetone and placed at −20 °C for 5 min, followed by a 5 min $H_2O$ wash. The larvae were then washed for 4×5 min in PBST. For all immunostains larvae were blocked overnight at 4 °C in blocking solution (2% goat serum, 1% bovine serum albumin, 2% fish skin gelatin in PBST or PBDTT). Larvae were then incubated in primary antibodies in antibody solution (1% bovine serum albumin in PBST or PBDTT) overnight, nutating at 4 °C. The next day, the larvae were washed for 4×5 min in PBST or PBDTT to remove the primary antibodies. Secondary antibodies in antibody solution were added and larvae were incubated for 2 hr at room temperature, with minimal exposure to light. Secondary antibodies were washed out with PBST or PBDTT for 4×5 min. Larvae were mounted on glass slides with Prolong Gold (ThermoFisher Scientific) using No. 1.5 coverslips. Primary antibodies used were:

> Rabbit anti-Myo7a (Proteus 25–6790; 1:1000)
> Mouse anti-Ribeye b (IgG2a) (*Sheets et al., 2011*)
> Mouse anti-pan-MAGUK (IgG1) (Millipore MABN7; 1:500)
> Mouse anti-Myo7a (DSHB 138–1; 1:500)
> Rabbit anti-Emx2 (Trans Genic KO609; 1:250)

The following secondaries were used at 1:1000 for synaptic labeling: goat anti-rabbit Alexa 488, goat anti-mouse IgG2a Alexa 546, goat anti-mouse IgG1 Alexa 647, along with Alexa 488 Phalloidin (Thermofischer; #A-11008, #A-21133, #A-21240, #A12379). For Emx2 co-labeling the following secondaries were used at 1:1000: goat anti-mouse Alexa 488, and goat ant-rabbit Alexa 647, along with Alexa 488 Phalloidin (Thermofischer; #A12379, #A28175, #A27040).

Fixed samples were imaged on an inverted Zeiss LSM 780 laser-scanning confocal microscope with an Airyscan attachment (Carl Zeiss AG, Oberkochen, Germany) using an 63×1.4 NA oil objective lens. Airyscan z-stacks were acquired every 0.18 μm. The Airyscan Z-stacks were processed with Zen

Black software v2.1 using 2D filter setting of 6.0. Experiments were imaged with the same acquisition settings to maintain consistency between comparisons. Processed imaged were further processed using Fiji.

## Zebrafish immunostain quantification (Figures 5, 9 and 10)

Images were processed in ImageJ. Researcher was not blinded to genotype. Hair bundle orientation was scored relative to the midline of the muscle somites. HC number per neuromast were quantified based on Myo7a labeling and presence of a paired/complete synapse. For quantification of Emx2 labeling, HCs were scored as Emx2 positive if they labeled with both Emx2 and Myo7a. To quality as a ribbon or presynapse, the following minimum size filters were applied to images: Ribeye b: $0.025 \, \mu m^2$, MAGUK: $0.04 \, \mu m^2$. A complete synapse was comprised of both a Ribeye b and MAGUK puntca. An unpaired presynapse consisted of only a Ribeye b puntca, while an unpaired postsynapse consisted of only a MAGUK puncta. In each neuromast all HCs (~15 HC per neuromast) were examined for our quantifications.

## Zebrafish functional calcium imaging in hair bundles (Figure 5)

GCaMP6s-based calcium imaging in zebrafish hair bundles has been previously described (*Lozano-Ortega et al., 2018*). Briefly, individual 5–6 dpf larvae were first anesthetized with tricaine (0.03% Ethyl 3-aminobenzoate methanesulfonate salt, SigmaAldrich). To restrain larvae, they were then pinned to a Sylgard-filled recording chamber. To suppress the movement, alpha-bungarotoxin (125 μM, Tocris) was injected into the heart. Larvae were then rinsed and immersed in extracellular imaging solution (in mM: 140 NaCl, 2 KCl, 2 CaCl$_2$, 1 MgCl$_2$ and 10 HEPES, pH 7.3, OSM 310+/-10) without tricaine. A fluid jet was used to mechanically stimulate the apical bundles of HCs of the A-P neuromasts of the primary posterior lateral-line. To stimulate the two orientations of HCs (A to P and P to A) a 500ms 'push' was delivered. Larvae were rotated 180° to deliver a comparable 'push' stimulus to both the A to P and P to A HCs.

To image calcium-dependent mechanosensitive responses in apical hair bundles, a Bruker Swept-field confocal system was used. The Bruker Swept-field confocal system was equipped with a Rolera EM-C2 CCD camera (QImaging) and a Nikon CFI Fluor 60×1.0 NA water immersion objective. Images were acquired in 5 planes along the Z-axis at 0.5 μm intervals (hair bundles) at a 50 Hz frame rate (10 Hz volume rate). The 5 plane Z-stacks were projected into one plane for image processing and quantification. The method to create spatial ΔF heatmaps has been described (*Lozano-Ortega et al., 2018*). For GCaMP6s measurements, a circular ROI with a~1.5 μm (hair bundles) diameter was placed on the center of each individual bundle. The mean intensity ($\Delta F/F_0$) within each ROI was quantified. $F_0$ represents the GCaMP6s intensity prior to stimulation. We examined the GCaMP6s signal in each hair bundle to determine its orientation. The GCaMP6s responses for each neuromast were averaged to quantify the magnitude of the A to P and P to A responses.

## Zebrafish sparse labeling of single afferents in the lateral-line (Figure 9)

To visualize the innervation pattern of single afferent neurons, a *neuroD1:tdTomato* plasmid was injected into zebrafish embryos at the one-cell stage. This plasmid consists of a 5 kb minimal promoter, *neurod1*, that drives tdTomato expression in lateral-line afferents (*Ji et al., 2018*). This plasmid was pressure at a concentration of 30 ng/μl. At 3 dpf larvae were anesthetized with 0.03% ethyl 3-aminobenzoate methanesulfonate (Tricaine), to screen for tdTomato expression. Larvae were screened for mosaic expression of tdTomato expression in the lateral-line afferents using a Zeiss SteREO Discovery V20 microscope (Carl Zeiss) with an X-Cite 120 external fluorescent light source (EXFO Photonic Solutions Inc). After selecting larvae with tdTomato expression, larvae were prepared for immunostaining at 5 dpf, and imaged as outlined above.

## Zebrafish data analysis and statistics

Statistical analyses and data plots were performed with Prism 9 (Graphpad, San Diego, CA). Values of data with error bars on graphs and in text are expressed as mean ± SEM unless indicated otherwise. All experiments were performed on a minimum of three animals, six neuromasts (primary posterior lateral-line neuromasts with A-P orientations: L1-L5). For 5–6 dpf larvae, each neuromast represents analysis from 12 to 20 HCs and 41–68 synapses. All replicates are biological. No animals or samples

were excluded from our analyses unless control experiments failed–in these cases all samples were excluded. No randomization or blinding was used for our animal studies. Where appropriate, data was confirmed for normality using a D'Agostino-Pearson normality test. Statistical significance between conditions was determined by an unpaired $t$-test.

## Acknowledgements

This work was supported by the National Institute on Deafness and Other Communication

Disorders (NIDCD) grant R01DC018304 (to RAE, KEC and BT), R01DC015242 (to BT) and NIDCD Intramural Research Program Grant 1ZIADC000085-01 (to KK).

## Additional information

### Funding

| Funder | Grant reference number | Author |
|---|---|---|
| National Institute on Deafness and Other Communication Disorders | R01 DC018304 | Ruth Anne Eatock Kathleen E Cullen Basile Tarchini |
| National Institute on Deafness and Other Communication Disorders | R01 DC015242 | Basile Tarchini |
| National Institute on Deafness and Other Communication Disorders | 1ZIADC000085-01 | Katie S Kindt |

The funders had no role in study design, data collection and interpretation, or the decision to submit the work for publication.

### Author contributions

Kazuya Ono, Amandine Jarysta, Natasha C Hughes, Alma Jukic, Data curation, Formal analysis, Validation, Investigation, Visualization, Methodology, Writing – review and editing; Hui Ho Vanessa Chang, Data curation, Formal analysis, Validation, Visualization, Methodology; Michael R Deans, Resources; Ruth Anne Eatock, Kathleen E Cullen, Katie S Kindt, Basile Tarchini, Conceptualization, Data curation, Formal analysis, Supervision, Funding acquisition, Validation, Investigation, Visualization, Methodology, Writing – original draft, Project administration, Writing – review and editing

### Author ORCIDs

Kazuya Ono ⓘ http://orcid.org/0000-0002-3857-0055
Amandine Jarysta ⓘ https://orcid.org/0000-0002-9519-3559
Natasha C Hughes ⓘ https://orcid.org/0000-0003-2031-3306
Hui Ho Vanessa Chang ⓘ http://orcid.org/0009-0005-9924-6143
Michael R Deans ⓘ https://orcid.org/0000-0001-6319-7945
Ruth Anne Eatock ⓘ https://orcid.org/0000-0001-7547-2051
Kathleen E Cullen ⓘ https://orcid.org/0000-0002-9348-0933
Katie S Kindt ⓘ https://orcid.org/0000-0002-1065-8215
Basile Tarchini ⓘ https://orcid.org/0000-0003-2708-6273

### Ethics

All mouse work was reviewed for compliance and approved by the Animal Care and Use Committees of The Jackson Laboratory (anatomical experiments), University of Chicago (cellular electrophysiology and afferent labeling experiments) and JohnsHopkins University School of Medicine (behavioral experiments). Zebrafish work was performed at the National Institute of Health (NIH) and approved by the Animal Use Committee at the NIH under animal study protocol #1362-13.

Reviewer #1 (Public review): https://doi.org/10.7554/eLife.97674.3.sa1
Reviewer #2 (Public Review): https://doi.org/10.7554/eLife.97674.3.sa2

Author response https://doi.org/10.7554/eLife.97674.3.sa3

## Additional files

### Supplementary files
- Supplementary file 1. Percent of LES HCs with transduction.
- Supplementary file 2. Genotype comparisons of transduction and adaptation in LES HCs. n.d., very fast component not detected. n/a, not available.
- Supplementary file 3. Genotype comparisons of electrical properties of Type I and II HCs in LES. *estimated power of non-significant result.
- Supplementary file 4. Genotype comparison of G(X) and adaptation parameters (all hair cells). Includes both cell types and all zones. n.d., very fast component not detected.
- Supplementary file 5. Genotype comparison of excitability in LES afferents. *estimated power of non-significant result.
- MDAR checklist

### Data availability
The data supporting findings in this study are available with the following identifier DOIs: https://doi.org/10.5061/dryad.905qfttvj (Eatock lab), https://doi.org/10.5061/dryad.m63xsj4br (Kindt lab), https://doi.org/10.5281/zenodo.13839293 (Tarchini and Cullen labs).

The following datasets were generated:

| Author(s) | Year | Dataset title | Dataset URL | Database and Identifier |
|---|---|---|---|---|
| Kindt K, Jukic A | 2024 | Data from: Contributions of mirror-image hair cell orientation to mouse otolith organ and zebrafish neuromast function | https://doi.org/10.5061/dryad.m63xsj4br | Dryad Digital Repository, 10.5061/dryad.m63xsj4br |
| Jarysta A, Tarchini B, Cullen KE, Hughes NC, Chang HHV | 2024 | Contributions of mirror-image hair cell orientation to mouse otolith organ and zebrafish neuromast function | https://doi.org/10.5281/zenodo.13839293 | Zenodo, 10.5281/zenodo.13839293 |
| Ono K, Eatock RA | 2024 | Contributions of mirror-image hair cell orientation to mouse otolith organ and zebrafish neuromast function | https://doi.org/10.5061/dryad.905qfttvj | Dryad Digital Repository, 10.5061/dryad.905qfttvj |

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
