## [Editor Report · eLife Assessment]

This **valuable** study provides **convincing** evidence that mutant hair cells with abnormal, reversed polarity of their hair bundles in mouse otolith organs retain wild-type localization, mechanoelectrical transduction and firing properties of their afferent innervation, leading to mild behavioral dysfunction. It thus demonstrates that the bimodal pattern of afferent nerve projections in this organ is not causally related to the bimodal distribution of hair-bundle orientations, as also confirmed in the zebrafish lateral line. The work will be of interest to scientists interested in the development and function of the vestibular system as well as in planar cell polarity.

---

## [Referee Report · Reviewer #1 (Public review)]

Summary:

The authors aim at dissecting the relationship between hair-cell directional mechanosensation and orientation-linked synaptic selectivity, using mice and the zebrafish. They find that Gpr156 mutant animals homogenize the orientation of hair cells without affecting the selectivity of afferent neurons, suggesting that hair-cell orientation is not the feature that determines synaptic selectivity. Therefore, the process of Emx2-dependent synaptic selectivity bifurcates downstream of Gpr156.

Strengths:

This is an interesting and solid paper. It solves an interesting problem and establishes a framework for the following studies. That is, to ask what are the putative targets of Emx2 that affect synaptic selectivity.

The quality of the data is generally excellent.

Weaknesses:

The feeling is that the advance derived from the results is very limited.

Comments on revised version:

I am happy with the authors' reply and do not wish to modify my initial assessment.

---

## [Referee Report · Reviewer #2 (Public Review)]

Summary:

The authors inquire in particular whether the receptor Gpr156, which is necessary for hair cells to reverse their polarities in the zebrafish lateral line and mammalian otolith organs downstream of the differential expression of the transcription factor Emx2, also controls the mechanosensitive properties of hair cells and ultimately an animal's behavior. This study thoroughly addresses the issue by analyzing the morphology, electrophysiological responses, and afferent connections of hair cells found in different regions of the mammalian utricle and the Ca2+ responses of lateral line neuromasts in both wild-type animals and gpr156 mutants. Although many features of hair cell function are preserved in the mutants-such as development of the mechanosensory organs and the Emx2-dependent, polarity-specific afferent wiring and synaptic pairing-there are a few key changes. In the zebrafish neuromast, the magnitude of responses of all hair cells to water flow resembles that of the wild-type hair cells that respond to flow arriving from the tail. These responses are larger than those observed in hair cells that are sensitive to flow arriving from the head and resemble effects previously observed in Emx2 mutants. The authors note that this behavior suggests that the Emx2-GPR156 signaling axis also impinges on hair cell mechanotransduction. Although mutant mice exhibit normal posture and balance, they display defects in swimming behavior. Moreover, their vestibulo-ocular reflexes are perturbed. The authors note that the gpr156 mutant is a good model to study the role of opposing hair cell polarity in the vestibular system, for the wiring patterns follow the expression patterns of Emx2, even though hair cells are all of the same polarity. This paper excels at describing the effects of gpr156 perturbation in mouse and zebrafish models and will be of interest to those studying the vestibular system, hair cell polarity, and the role of inner-ear organs in animal behavior.

The study is exceptional in including, not only morphological and immunohistochemical indices of cellular identity but also electrophysiological properties. The mutant hair cells of murine maculæ display essentially normal mechanoelectrical transduction and adaptation-with two or even three kinetic components-as well as normal voltage-activated ionic currents.

---

## [Author Response]

The following is the authors’ response to the original reviews.

**Recommendations for the authors:**
- The authors should think about revising the terminology used to describe electrophysiological data in zebrafish (Fig.5): "posterior" hair cells in a neuromast are sensitive to posterior-to-anterior flow, which is currently termed "anterior". This is confusing because when "posterior" or "anterior" is used, for instance in the labels of the figure, one may get confused about whether this applies to hair-cell position or directionality of the stimulus. It would help to always use clearer terminology for the stimulus (e.g. posterior-to-anterior (P-to-A) as in Kindig 2023, or "from the tail"). Also, the authors may want to clarify what we should see in Fig.5 demonstrating that posterior hair cells, with reversed hair-bundle polarity, actually evince transduction of similar magnitude as anterior hair cells, with normal polarity of their hair bundles.

This nomenclature can indeed be confusing. Per the reviewers request we have changed the terminology to always refer to the direction of flow sensed by the hair cells. For example, HCs that respond to posterior-directed flow or anterior-directed flow. We now denote these HCs as (A to P) and (P to A), respectively in the Figure for clarity. We have modified Figure 5, the Figure 5 legend and Results (starting line 339) to reflect these changes.

In addition, in our results we now provide more context when comparing the response magnitude of the anterior-sensing hair cells in gpr156 mutants to the response magnitude of the two diVerent orientations of hair cells in controls.

- Also, does it make sense that there is no defect in MET for mouse otolith organs with deleted GPR156, whereas there is a diVerence in the zebrafish lateral line? It would help motivate the study on mechanoelectrical transduction (see comment of Reviewer 1 below).

We previously discussed this point and recognized that subtle eVects remain possible in mouse (previously Discussion line 614). We have now modified the text in the Discussion to better emphasize this point (new line 627). The Eatock lab is currently working on developing calcium imaging in the mouse utricle to revisit this question in a future study. "Subtle eects remain possible, however, given the variance in single-cell electrophysiological data from both control and mutant mice. Nevertheless, current results are consistent with normal HC function in the Gpr156 mouse mutant, a prerequisite to interrogate how non-reversed HCs aects vestibular behavior."

To help motivate transduction studies starting in the second Result paragraph, we added a transition at Line 205 that was indeed lacking:

"Gpr156 inactivation could be a powerful model to specifically ask how HC reversal contributes to vestibular function. However, GPR156 may have other confounding roles in HCs besides regulating their orientation, similar to EMX2, which impacts mechanotransduction in zebrafish HCs (Kindig et al., 2023) and aerent innervation in mouse and zebrafish HCs (Ji et al., 2022; Ji et al., 2018)."

(1) One overarching objective of this study was to use the Gpr156 KO model to discover how polarity reversal informs vestibular function (Introduction, overall summary in the last paragraph) . Pairing behavioral defects with hair cell orientation is only possible if hair cell transduction is normal, which had to be tested.

(2) The notion that experiments that produced negative results are unecessary and are not properly motivated can only apply in retrospect. At early stages we performed electrophysiology because we did not know whether transduction would be normal in absence of GPR156. We also did not know whether innervation would be normal. The fact that both appear normal makes Gpr156 KO a better model to address the importance of orientation reversal (conclusion of the Discussion line 705).

See also reply to Reviewer #1 below.

**Reviewer #1 (Recommendations For The Authors):**
Fig1, panel B appears to show diVerent focal planes for Gpr156del/+ and Gpr156del/del.

Figure 1B had control and mutant panels at slightly diVerent focal planes indeed. We swapped the right (mutant) panel image and adjusted intensities in the control image to match adjustments of the new mutant image.

Given that this work is largely about polarity and connectivity to neurons, I do not understand the need to assess mechanosensitivity in Gpr156 mutants. Please explain in the text, as follows: "After establishing normal numbers and types of mouse vestibular HCs, we assessed whether HCs respond normally to hair bundle deflections in the absence of GPR156." We did this because...

Please see reply above in 'Recommendations for the authors' for comment about the need to assess mechanosensitivity. We agree that this transition was lacking, and we added an explanation as recommended:

"Gpr156 inactivation could be a powerful model to specifically ask how HC reversal contributes to vestibular function. However, GPR156 may have other confounding roles in HCs besides regulating their orientation, similar to EMX2, which impacts mechanotransduction in zebrafish HCs (Kindig et al., 2023) and aerent innervation in mouse and zebrafish HCs (Ji et al., 2022; Ji et al., 2018)."

Anyway, the data in Figures 2, 3 and 4 seems somewhat superfluous to the main message of the paper.

Please see reply above in 'Recommendations for the authors'. This data may appear superfluous in retrospect but we could not claim that behavioral changes in Gpr156 mutants reflect the role of the line of polarity reversal if, for example, hair cell transduction was abnormal. We had to perform experiments to figure this out. We were further motivated as data began to emerge from the zebrafish lateral line that showed eVects on HC transduction. Although we did not get positive results on this question in the mouse, we think the diVerence between models should be included as a significant part of the narrative.